# Cloud Condensation Nuclei Activity of CaCO$_3$ Particles with Oleic Acid and Malonic Acid Coatings

**Mingjin Wang[1,2], Tong Zhu[1*], Defeng Zhao[2], Florian Rubach[2,3], Andreas Wahner[2], Astrid Kiendler-Scharr[2], and Thomas F. Mentel[2*]**

[1]BIC-ESAT and SKL-ESPCl, College of Environmental Sciences and Engineering, Peking University, Beijing 100871, China.

[2]Institut für Energie- and Klimaforschung (IEK-8), Forschungszentrum Jülich GmbH, 52425 Jülich, Germany.

[3]Klimageochemie, Max Planck Institut für Chemie, 55128 Mainz, Germany

[*] To whom correspondence should be addressed: tzhu@pku.edu.cn; t.mentel@fz-juelich.de

**Abstract.**

Condensation of carboxylic acids on mineral particles will lead to coatings, and impact on the particles' potential to act as cloud condensation nuclei (CCN). To determine how the CCN activity of mineral particles is impacted by carboxylic acid coatings, the CCN activities of CaCO$_3$ particles and CaCO$_3$ particles with oleic acid and malonic acid coatings were compared in this study. The results revealed that small

amounts of oleic acid coating (volume fraction (vf) $\leq$ 4.3%) decreased the CCN
activity of $CaCO_3$ particles, while more oleic acid coating (vf $\geq$16%) increased the
CCN activity of $CaCO_3$ particles. This phenomenon has not been reported before. On
the other hand, the CCN activity of $CaCO_3$ particles coated with malonic acid
increased with the thickness of the malonic acid coating (vf = 0.4 - 40%). Even
smallest amounts of malonic acid coating (vf = 0.4%) significantly enhanced the CCN
activity of $CaCO_3$ particles from $\kappa$ = 0.0028 $\pm$ 0.0001 to $\kappa$ = 0.0123 $\pm$ 0.0005. This
supports that a small amount of water-soluble organic acid coating may significantly
enhance the CCN activity of mineral particles. The presence of water vapor during the
coating process with malonic acid additionally increased the CCN activity of the
coated $CaCO_3$ particles, probably because more $CaCO_3$ reacts with malonic acid if
sufficient water is available.

## 1 Introduction

Atmospheric aerosols serve as cloud condensation nuclei and change the radiative properties (cloud albedo effect) and lifetime (cloud lifetime effect) of clouds, thus affecting the Earth's climate indirectly (Liu and Wang, 2010; Gantt et al., 2012; Penner et al., 2004; Haywood and Boucher, 2000). Mineral aerosol is one of the most abundant components of the atmospheric aerosol. It is estimated that 1500-2600 Tg of mineral aerosol particles with radii between 0.1 and 8 μm are emitted annually into the atmosphere on a global scale (Cakmur et al., 2006). Mineral aerosol particles are mainly composed of substances that are slightly soluble or insoluble in water. Cloud condensation nuclei (CCN) activity measurements show that the hygroscopicity parameter κ (Petters and Kreidenweis, 2007) varies between 0.001 and 0.08 for mineral aerosols, including $CaCO_3$ aerosol, clay aerosols and mineral dust aerosols generated in the laboratory or sampled from various locations worldwide (Garimella et al., 2014; Yamashita et al., 2011; Zhao et al., 2010; Koehler et al., 2009; Sullivan et al., 2010; Herich et al., 2009). The low κ indicates that the CCN activity of mineral aerosol is much lower than that of water soluble salts like $(NH_4)_2SO_4$ (κ = 0.61) and NaCl (κ = 1.28), which are also common in atmospheric aerosols (Petters and Kreidenweis, 2007). Tang et al. (2016) reviewed recently the interaction of mineral dust particles with water.

Mineral aerosol particles can be coated by organic vapors during their residence and transport in the atmosphere. Many individual particle measurements have shown that mineral components and organic matter can coexist in the same individual aerosol

particle in the real atmosphere (Falkovich et al., 2004; Falkovich et al., 2001; Russell
et al., 2002; Li and Shao, 2010). Carboxylic acids (R(C=O)OH) are abundant species
among the organic matter that coexists with mineral particles. Russell et al. (2002)
found that carbonyls (R(C=O)R), alkanes,   and R(C=O)OH are present in individual
mineral (and sea salt) aerosol particles, with enhanced concentration of R(C=O)OH.
They also found that $Ca^{2+}$, $CO_3^{2-}$, R(C=O)OH and R(C=O)R coexisted in some
individual mineral aerosol particles with a strong correlation between $CO_3^{2-}$ and
R(C=O)OH. These particles could be formed by $CaCO_3$ particles (partly) coated with
organic film. Falkovich et al. (2004) also found that organic and inorganic
components coexisted in individual mineral aerosol particles with the organic
component consisting of various short-chain ($C_1$-$C_{10}$) mono- and dicarboxylic acids
(MCA and DCA). The concentration of short-chain carboxylic acids in mineral
aerosol particles increased with the increase of the ambient relative humidity. A
possible explanation for such observations could be that when more water is
condensed onto mineral particles at higher ambient relative humidity, the adsorbed
carboxylic acids are ionized in the aqueous environment and react with mineral
particles forming organic acid salts. Of the major components of mineral aerosol
particles (clay, calcite ($CaCO_3$), quartz, mica, feldspar, etc.), only $CaCO_3$ with
alkaline character can react with carboxylic acids in this way. Thus $CaCO_3$ may play a
key role in the uptake of carboxylic acids by mineral aerosol particles.
Carboxylic acid coatings on mineral aerosol particles change their chemical
composition and thus may have an impact on their CCN activity. Many previous
studies have investigated the CCN activity of pure mineral aerosol (Garimella et al.,
2014; Yamashita et al., 2011; Zhao et al., 2010; Koehler et al., 2009; Sullivan et al.,
2010; Herich et al., 2009) and pure carboxylic acid aerosol (Kumar et al., 2003; Hori
et al., 2003; Cruz and Pandis, 1997; Hartz et al., 2006), but only a few studies have
investigated the CCN activity of mineral aerosol particles with carboxylic acid
coatings (Tang et al., 2015; Hatch et al., 2008; Gierlus et al., 2012).
In this study we used malonic acid and oleic acid as coating materials and $CaCO_3$
particles as cores, and investigated the CCN activity of the coated $CaCO_3$ particles.
Herein we varied the coating thickness and the relative humidity during the coating
process. Malonic acid is a representative of the class of dicarboxylic acids and oleic
acid is an example of surfactant like compounds. Dicarboxylic acids are ubiquitous in
the atmosphere (Kawamura et al., 1996; Kawamura and Ikushima, 1993; Ho et al.,
2007; Mkoma and Kawamura, 2013; Kawamura and Bikkina, 2016) and formed by
photochemical reactions and ozonolysis (Chebbi and Carlier 1996; Kawamura and
Bikkina, 2016; Kawamura et al., 1996; Khare et al. 1999; Mellouki et al., 2015). It has
been reported that dicarboxylic acids ($C_2$-$C_{10}$) account for 0.06-1.1% of the total
aerosol mass, with higher values in the summer, and 1.8% of the total aerosol carbon
(TC) in urban aerosol, in which oxalic acid, malonic acid, and succinic acid are the
most abundant species (Kawamura et al., 1996; Kawamura and Ikushima, 1993; Ho et
al., 2007; Mkoma and Kawamura, 2013). Oleic acid, which is emitted into the
atmosphere by the cooking of meat, wood burning, and automobile source (Schauer et
al., 1999; Rogge et al., 1998; Rogge et al., 1993), is present in atmospheric aerosols of
urban, rural, and forest areas (Cheng et al., 2004; Ho et al., 2010). The water
solubility of the two organic acids is complementary; it is high for malonic acid while
it is very low for oleic acid. Coatings of malonic acid and oleic acid could thus have
different effects on CCN activity of mineral particles.


## 2 Experimental

As general procedure, $CaCO_3$ aerosol was generated according to Zhao et al. (2010), and then poly- or monodisperse $CaCO_3$ aerosol particles were coated by malonic or oleic acid in a coating device. A flow tube was optionally applied to extend the residence time. The particle size, chemical composition, and CCN activity of the $CaCO_3$ particles were measured before and after coating. Figure 1 shows the schematic of the experimental set up.

### 2.1 Generation of $CaCO_3$ aerosol

$CaCO_3$ aerosol was generated by spraying a saturated $Ca(HCO_3)_2$ solution. A sample of $CaCO_3$ powder (2 g, pro analysis, $\geq$99%, Merck, Darmstadt, Germany) was suspended in 1-L Milli-Q water (18.2 M$\Omega$cm, TOC <5 ppb). Then about 1.5 L min$^{-1}$ $CO_2$ (purity $\geq$99.995%, Praxair Industriegase GmbH & Co. KG, Magdeburg, Germany) was bubbled into the suspension at room temperature for 3 h, while the suspension was stirred using a magnetic stirrer. During bubbling, $CO_2$ reacted with $CaCO_3$ to produce $Ca(HCO_3)_2$. After bubbling, the suspension was allowed to settle for 10 min, the supernatant clear $Ca(HCO_3)_2$ solution was decanted and used for spraying by a constant output atomizer (Model 3076, TSI, Shoreview, MN, USA) using 1.75 L min$^{-1}$ high-purity $N_2$ (Linde LiPur 6.0, purity 99.9999%, Linde AG, Munich, Germany).


The major portion (0.9 L min$^{-1}$) of the aerosol flow generated by spraying was dried
in a diffusion drier filled with silica gel. The relative humidity was below 10% after
drying. The remainder of the aerosol flow was drawn off by a pump and discarded.
The dry aerosol was passed through a tube furnace (Model RS 120/1000/12,
Nabertherm GmbH, Lilienthal, Germany) set at 300 ºC. The residence time of the
aerosol in the furnace was 5.2 s. Zhao et al. (2010) described this method for
generating $CaCO_3$ aerosol in detail. At room temperature dry $Ca(HCO_3)_2$ is
thermodynamically unstable and decays into $CaCO_3$, $CO_2$, and $H_2O$ (Keiser and
Leavitt, 1908). With this method the aerosol still contained some $Ca(HCO_3)_2$ after
drying, but after heating at 300 ºC it was completely converted into $CaCO_3$ (Zhao et
al., 2010). The $CaCO_3$ aerosol generated was either first size selected by a Differential
Mobility Analyzer (DMA, TSI 3081) with a neutralizer on the inlet or entered the
coating device directly as poly-disperse aerosol.
Figure 2 (upper panel) shows the total number concentration and mean size of the
bare $CaCO_3$ aerosol particles generated at different spraying time, which were
measured with the SMPS described below. The mean size stabilized after about 50
min in the range 49.8-55.5 nm. Over the 232 min spraying time, the total number
concentration varied in the range $1.8 \cdot 10^6$–$4.5 \cdot 10^6$ cm$^{-3}$. The total number
concentration decreased by about 1/3 in the initial 70 min. The decrease became
slower after 70 min and the total number concentration tended to stabilize after 155
min. After 70 min the total number concentration varied in a smaller range of $1.8 \cdot 10^6$–
$2.9 \cdot 10^6$ cm$^{-3}$, therefore, the measurements in this study typically started after 70 min
spraying. The typical size distribution of the $CaCO_3$ aerosol particles after 70 min
spraying is shown in Fig. 2 (lower panel). The $CaCO_3$ particles showed a single mode
distribution with a mode diameter at 32.2 nm. The number concentration was more
than 10000 $cm^{-3}$ for particles between 12.6 and 151.2 nm.
**2.2 Organic acid coating**
The coating device (Fig. 1, right hand side) used in this study was designed by Roselli
(2006), and showed good reproducibility, controllability, and stability. The glass
apparatus consisted of a small storage bulb (100 ml) holding the organic coating
substances which was directly connected to a mixing cell (about 35 ml). The storage
bulb and mixing cell were fully immersed in a flow-through water heater connected to
a thermostatic bath (F25, Julabo GmbH, Seelbach, Germany). The temperature range
of the thermostatic bath used in this study was 30-80 ºC. An extra $N_2$ stream could be
passed through the storage bulb in order to enhance the organic vapors flowing into
the mixing cell. The outflow of the coating device was connected to a Liebig type
water cooler. The water cooler was controlled by another thermostatic bath (F25,
Julabo GmbH, Seelbach, Germany) operated at 25 ºC throughout all of the
experiments.
The bottom of the storage bulb was filled with either 5.0 g malonic acid powder
(assay ≥98%(T), Fluka Chemika, Sigma-Aldrich, St Louis, MO, USA) or 10.0 ml
oleic acid (chemical purity (GC) 99.5%, Alfa Aesar, Ward Hill, MA, USA). A flow of
0.9 L $min^{-1}$ high purity $N_2$ was used to carry the organic acid vapor up into the mixing
cell. The flow of 0.9 L $min^{-1}$ $CaCO_3$ aerosol was passed through the mixing cell and
mixed with the 0.9 L $min^{-1}$ $N_2$ flow carrying the organic acid vapor. The mixed flow
then entered the water cooler. The organic acid vapor was condensed on $CaCO_3$
aerosol particles in both the mixing cell and the water cooler. The residence time of
the aerosol in the coating device including the cooler was about 6 seconds Three
identical coating devices, with the same heating and cooling thermostatic bath, were
used: one for malonic acid coating, one for oleic acid coating, and a blank one without
organic acid for assessing the impact caused by heating the $CaCO_3$ aerosol in the
coating device without organic acid (Roselli, 2006).
The aerosol could enter the measuring instruments directly, or after passing through a
flow tube to increase its residence time. The flow tube was made of a straight circular
glass tube with a 2.5 cm internal diameter. The aerosol flow in the flow tube was
laminar flow. The average residence time of the aerosol in the flow tube was 23.7 s.
For the coating process we mixed flows of 0.9 L min$^{-1}$ of dry $N_2$ and of aerosol dried
to <10% relative humidity (RH) at room temperature (RT). As a consequence RH at
the outlet of the coating device was <5% at RT. To investigate the impact of water on
the coating process and CCN activity, organic coating at a higher relative humidity
was also performed. For that a bubbling device filled with Milli-Q water was utilized
to saturate the $N_2$ stream with water vapor before it entered the storage bulb (RH>90%
at RT). After mixing with the aerosol stream at RH <10%, the water concentration in
the mixing cell corresponded to RH ≈ 50% at RT or a partial pressure of ≈1500 Pa.
The relative humidity of the aerosol at the outlet of the coating device at RT was
indeed ~47% when humidification was applied. For the partial water vapor pressure
of 1500 Pa we calculated RH >7% at 60°C (for MA), and RH >3% at 80°C (for OA)
which is about an order of magnitude higher than RH in the dry cases. In fact RH will
be somewhat higher as the gas-phase may not reach the bath temperature which
primarily serves to warm up the coating agent and control its vapor pressure.

**2.3 Size and chemical composition measurements**

The number size distribution of the aerosol particles was measured using a Scanning
Mobility Particle Sizer (SMPS, TSI 3080 Electrostatic Classifier with TSI 3081 DMA,
TSI 3786 UWCPC). The sample flow was set to 0.6 L min$^{-1}$ and the sheath flow was
set to 6.0 L min$^{-1}$. The size range measured was 9.82-414.2 nm with a size resolution
of 64 channels per decade and the time resolution was 3 min for a complete scan.
Despite the maximum resolution of the SMPS the size bin width was still substantial
compared to the observed growth by coating. We therefore derived the diameter of the
coated (and the respective bare $CaCO_3$ particles) by interpolating in between the size
bins. For that we considered 5-9 size bins around the size bin of nominal mode and
fitted a lognormal distribution to these data. The fitted mode positions are listed in
Table 1. The error bars in x direction in Figure S1 in the supplement, show the shifts
of the fitted mode position relative to the nominal size bin.
The chemical composition and the vacuum aerodynamic diameter of the aerosol
particles were measured using a High-Resolution Time-of-Flight Aerosol Mass
Spectrometer (HR-ToF-AMS, Aerodyne Research Inc., Billerica, MA, USA (DeCarlo
et al., 2006)). The aerosol particles were vaporized at 600 ºC and ionized by electron
impact ionization at 70 eV, i.e. we focused on the measurements of the organic
coatings and sacrificed a direct $CaCO_3$ determination by AMS (compare Zhao et al.
2010). The AMS was routinely operated in V-mode in two alternating modes: 1 min
MS mode to measure the chemical composition and 2 min PToF mode. Only MS
mode data were analyzed. AMS measurements and SMPS measurements were
synchronous and both were repeated at least four times for each sample. Size
information for bare $CaCO_3$ was taken from SMPS data in the blank coating device.
We used specific marker $m/z$ to derive the amount of organic coating. For pure oleic
acid the signal at $m/z41$ ($C_3H_5^+$) was reported to be the strongest signal in the mass
spectrum measured by Q-AMS at EI energy of 70 eV and vaporizer temperature of
600 ºC (Sage et al., 2009). The signal at $m/z41$ was also strongest for oleic acid
coatings in our HR mass spectra. In order to get a high signal to noise ratio we choose
the signal at $m/z41$ in the MS mode of the AMS measurement as a marker for oleic
acid in the coated $CaCO_3$ particles. There was no significant signal at $m/z41$ for the
uncoated $CaCO_3$ particles. The average background signal at $m/z41$ per single aerosol
particle corresponded to $2.7\pm0.9\cdot10^{-12}$ μg for bare $CaCO_3$. The average value
presented the baseline of the mass spectra and the standard deviation was derived
from the noise of the mass spectra at $m/z41$. Similarly, the signal at $m/z42$ ($C_2H_2O^+$)
was one of the strongest signals in the mass spectrum of pure malonic particles
measured by Q-AMS with EI energy of 70 eV and vaporizer temperature of 600 ºC
(Takegawa et al., 2007). That signal was also observed for malonic acid coatings in
our HR mass spectra and used as marker for malonic acid coatings. The average
background signal per aerosol particle at $m/z42$ for bare $CaCO_3$ particles was
$1.4\pm0.4\cdot10^{-12}$μg. The average value represented the baseline of the mass spectra at
$m/z42$ and the standard deviation was derived from the noise in the mass spectra.
The coating amount for both organic compounds was derived as follows. The
observed signal at the respective marker $m/z$ was corrected for the background signal
from bare $CaCO_3$ and then scaled to the volume increase (per particle) calculated
from the shift of the particle diameter $D_P$ for the largest coating amount achieved at 80
ºC coating temperature. Because of the relative large bin width compared to the
growth by coating we used the $D_P$'s, interpolated between the nominal size bins of the
SMPS (see above). This assumed spherical core shell morphology, based on Zhao et
al. (2011) where we showed that the $CaCO_3$ particles generated by our spray drying
method are spherical. The relation between AMS derived organic mass (baseline
corrected marker signals at *m/z41* or *m/z42* per particle) and SMPS derived organic
mass ($\rho_{org} \times \pi/6 \times (D_P - D_{PCaCO3})^3$) is linear within the limits of the method (see Figure
S1 in the supplement). For discussion we will refer to the AMS results, as we are able
to detect amounts of organic coatings as small as few time $10^{-12}$ µg per particle with
the AMS, while these could be not be detected by the SMPS.

**2.4 CCN activity measurement**

The aerosol was dried to RH <3% by another diffusion drier before the CCN activity
was measured. To determine the CCN activity of the aerosol, the number
concentration of the cloud condensation nuclei (CCN) of the aerosol was measured
with a continuous flow CCN counter (CCNC, DMT-100, Droplet Measurement
Technologies, Boulder, CO, USA). The total number concentration (CN) of the
aerosol particles was synchronously measured using an ultrafine water-based
condensation particle counter (UWCPC, TSI 3786, cf. Zhao et al., 2010). The ratio of
CCN to CN (CCN/CN) is called the activated fraction ($a_f$). In cases where a
poly-disperse aerosol was coated, the coated aerosol particles were size selected by
scanning a DMA between 10.6 and 478.3 nm , and the CCN and CN concentrations
were determined for each size bin while the supersaturation (SS) kept constant
(known as 'Scanning Mobility CCN Analysis (SMCA)', Moore et al., 2010). The
activated fraction was calculated after the CCN and CN concentrations were corrected
for the multiple charged particles.
The activated fraction as a function of the particle size was fitted with a cumulative
Gaussian distribution function (Rose et al., 2008). The turning point of the function is
the critical dry diameter ($D_{crit}$ or $D_{50}$) at the set SS. The activation efficiency (i.e., the
activated fraction when aerosol particles are completely activated) was 83% for the
CCN instrument, determined using 150 nm $(NH_4)_2SO_4$ particles at SS=0.85%.
Besides $CaCO_3$ and coated $CaCO_3$ particles, the CCN activities of malonic acid
particles, oleic acid particles, and mixed particles of $CaCO_3$ and malonic acid were
also measured. The malonic acid particles were generated by spray-drying a malonic
acid solution. The oleic acid particles were generated by heating 10.0 ml oleic acid to
97 ºC in the storage bulb and then cooling the vapor to 2 ºC in the water cooler in a
clean coating device. 1.75 L $min^{-1}$ high-purity $N_2$ was used as carrying gas and went
into the storage bulb through'1 $N_2$ in' entrance in Fig. 1; the'3 Aerosol in' entrance in
Fig. 1 was closed. This way, pure oleic acid particles with diameters up to 333 nm
were generated. Mixed $CaCO_3$/malonic acid particles were generated by spraying the
supernatant clear solutions which were prepared by settling suspensions containing
$CaCO_3$ and malonic acid in molar ratios of about 1:1 and 3:1. The suspensions were
prepared with 0.020 g malonic acid and 0.021 g $CaCO_3$ and 0.025 g malonic acid and
0.076 g $CaCO_3$ in 1000 ml Milli-Q water, respectively. The suspensions were allowed
to stand for 24h.
For aerosols where monodisperse aerosol particles with a dry diameter $D_P$ were
coated, the CCN concentration was measured at different SS and the CN
concentration was measured synchronously. Similarly, the activated fraction as a
function of SS was fitted with a sigmoidal function. The turning point of the function
is the critical supersaturation ($SS_{crit}$) and the corresponding dry diameter $D_P$ is called
the critical diameter, $D_{crit}$. The hygroscopicity parameter $\kappa$ (Petters and Kreidenweis,
2007) was then calculated from the $D_P(D_{crit})$-$SS_{crit}$ or $SS(SS_{crit})$-$D_{crit}$ data set. The SS
settings of the CCN counter were calibrated weekly using $(NH_4)_2SO_4$ aerosol based
on the theoretic values in the literature (summarized by Rose et al., 2008).
**3 Results and discussion**
**3.1 CCN activity of CaCO$_3$ aerosol**
Before the coating experiments we determined the CCN activity of the bare CaCO$_3$
aerosol particles. It was measured by the SMCA method using poly-disperse CaCO$_3$
aerosol particles. The value of the hygroscopicity parameter κ of the CaCO$_3$ aerosol
was $0.0028 \pm 0.0001$ derived by the least-square-fitting of $D_{crit}$ as a function of SS
($SS_{crit}$). This κ value is quite small, indicating that the CCN activity of the CaCO$_3$
aerosol is low. Our κ is well within the range of κ's of $0.0011 \pm 0.0004$ to $0.0070 \pm$
$0.0017$ found in previous studies for wet generated CaCO$_3$ particles (Zhao et al., 2010;
Sullivan et al., 2009; Gierlus et al., 2012, Tang et al., 2016), but larger than κ for dry
generated CaCO$_3$ aerosols (0.0008-0.0018, Sullivan et al., 2009).
The CCN activity for CaCO$_3$ aerosol passed through the blank coating device exposed
to temperatures of 60 ℃ and 80 ℃ was determined using the same method. The κ
value remained at $0.0028 \pm 0.0001$ up to 60 ℃ and increased to $0.0036 \pm 0.0001$ at 80
℃. The increase in κ by 0.0008 at 80 ℃ was smaller than the differences of reported κ
values for CaCO$_3$ aerosol in various studies, and much smaller than the changes of κ
values measured in this study when the CaCO$_3$ aerosol particles were coated by
malonic or oleic acid. So the effect of heating the CaCO$_3$ aerosol during the coating
process on the CCN activity of the $CaCO_3$ aerosol was neglected. The $D_{crit}$ at different
supersaturations ($SS_{crit}$) for the $CaCO_3$ aerosol and for the $CaCO_3$ aerosol passed
through a blank coating device at heating temperatures of 60 ºC and 80 ºC are shown
in Fig. 5 (red, yellow and green circles).
As the solubility of $CaCO_3$ in water is very low, droplet activation of $CaCO_3$ (and
other mineral dust components) is often described by a water adsorption approach,
wherein the solute term B in the Köhler equation (Köhler 1936, Seinfeld and Pandis,
2006, see eq. (S1-S3) in the supplement) is replaced by a water adsorption term. The
equations (1) and (2) show application of the Frenkel Halsey Hill adsorption isoterme
(FHH) as proposed by Sorjamaa and Laaksonen (2007) and Kumar et al. (2009):
$B = -A_{FHH} \cdot \theta^{-B_{FHH}}$                                         (1)
Therein the water coverage $\theta$ (Sorjamaa and Laaksonen, 2007) is given as:
$\theta = \dfrac{D_w - d_u}{2 \cdot 2.75 \cdot 10^{-4}}$     [µm]                    (2)
and $D_w$ and $d_u$ are the diameter of the wet particles and the insoluble core. We applied
the FHH parameter for $CaCO_3$ ($A_{FHH}$=0.25 and $B_{FHH}$=1.19, Kumar et al. 2009) and
derived a critical supersaturation of 1.52% for $CaCO_3$ particles with $d_u$ = 101.9 nm
(Figure 7, blue line). In comparison κ-Koehler theory predicts $SS_{crit}$=1.49% for
κ=0.0028. Such an $SS_{crit}$=1.49% would also be achieved by $8.5 \cdot 10^{-20}$ mole solute per
particle (Figure 7, black line). Figure 7 also shows the $SS_{crit}$ for the bare $CaCO_3$
particles processed at 80 °C temperature and the range of $SS_{crit}$ for 101.9 nm particles
calculated from the range of κ's given in the literature (Tang et al., 2016 and
references therein) for wet generated $CaCO_3$ particles.
We conclude that the surface of our $CaCO_3$ particles is a little more wettable than the
dry generated particles studied by Kumar et al. (2009). We presume formation of
$Ca(OH)(HCO_3)$ structures on the surface during the spray-drying generation process
as commonly observed whenever the $CaCO_3$ surface has been exposed to gaseous
water or liquid water (Stipp, 1999; Stipp and Hochella, 1991; Neagle and Rochester,
1990). In case of soluble components cauisng the lower $SS_{crit}$ their amount must be of
the order of $1 \cdot 10^{-19}$ mole per particles.

**3.2 CCN activity of $CaCO_3$ particles with oleic acid coating**

For the coating with oleic acid, we selected monodisperse $CaCO_3$ aerosol particles of
101.8 nm diameter using the DMA, and measured the size and chemical composition
of the particles before (uncoated) and after (coated) coating with oleic acid. The
results are listed in the upper part of Table 1.
The mode diameters of number size distribution for the uncoated $CaCO_3$ particles at
30-80 ºC remained in the 101.8 nm size bin, identical to that selected by the DMA.
Interpolation in between the size bins as described in the experimental section led to
an average dry diameters of bare $CaCO_3$ of $d_u$ = 101.9 nm. The mode diameters of the
$CaCO_3$ particles after coating with oleic acid in the range of 30-50 ºC stayed in the
pre-selected size bin at 101.8 nm, which means that the layers were too thin to
effectively grow the particles to the next size bin; the mode diameters increased
distinctively in the temperature range of 60-80 ºC (Fig. 3, upper panel). However, the
bin-interpolated diameters $D_P$ which are shown in Table 1 increased monotonically
over the whole temperature range.
The values of $m/z$41 [µg per particle] originating from the oleic acid coating for the
coated $CaCO_3$ particles at 30-80 ºC were at all temperatures significantly larger than
for the bare $CaCO_3$ particles, and increased with the increasing coating temperature
$(3.7 \cdot 10^{-12}$ - $390 \cdot 10^{-12}$ µg per particle, compare Table 1 and Fig. 3, bottom panel, red
circles). The AMS detected increase at $m/z41$ showed that the $CaCO_3$ particles already
contained small amounts of oleic acid after coating with oleic acid at temperatures
below 60 ºC although the $D_P$ shifted less than a size bin.
The organic volume fraction (vf) in the aerosol particles, $V_{OA}/V_{par}$ [%], was calculated.
Herein $V_{par} = (V_{OA} + V_{CaCO3})$, $V_{OA}$ is the oleic acid volume derived by AMS and
$V_{CaCO3}$ the volume of the bare $CaCO_3$ before coating (101.9 nm). $V_{OA}/V_{par}$ for the
uncoated $CaCO_3$ particles is by definition zero. The vf for the coated $CaCO_3$ particles
at 30-80 ºC increased with the increase in the coating temperature from 0.8% at 30 ºC
to 44% at 80 ºC (Fig. 3, bottom panel, green square and Table 1). The $CaCO_3$ particles
were indeed coated with a significant amount of oleic acid and the amount of oleic
acid coating increased with the increase in the coating temperature. The experiments
were repeated at least four times. The according standard deviations for the oleic acid
mass per particle in Table 1 demonstrate that the reproducibility of the experiments
was good and the performance of the coating device was stable.
The activated fractions at different SS for monodisperse $CaCO_3$ particles with $d_u =$
101.9 nm before and after oleic acid coating at 30-80 ºC are shown in Fig. 4. The top
panel in Fig. 4 shows the results at 30-60 ºC with up to $23\pm1.2\cdot10^{-12}$ μg of coating
material deposited on the $CaCO_3$ particles (vf up to 4.3%). At the lowest SS of 0.17%
and 0.35%, the activated fractions were very low and independent of the presence of
the coating material within the errors. When the SS increased to 0.52%, 0.70%, and
0.87%, the activated fractions for the coated $CaCO_3$ particles were *lower* than those
for the uncoated particles. Notably the activated fractions for the coated $CaCO_3$
particles *decreased* with the increase in the coating material in the range of vf
0.8-2.7%. The activated fractions for the $CaCO_3$ particles with different amounts of
coating spread with larger SS applied. However this trend reversed at the coating
temperature of 60 ºC and an oleic acid vf of 4.3%, and the activated fractions at vf =
4.3% became higher than those at 2.7% at the three largest SS. In summary, we found
that the CCN activity of the coated $CaCO_3$ particles with vf of OA in a range 0.8-4.3%
was lower than that of the uncoated $CaCO_3$ particles. The CCN activity of the coated
$CaCO_3$ particles decreased with the increasing vf in between 0.8-2.7%, i.e. the CCN
activity became lower when more coating material deposited on the $CaCO_3$ particles.
This trend turned at a vf somewhere between 2.7 and 4.3%. As the $D_P$ also increased
at 60 ºC we cannot differentiate if the increase in the activated fractions is due to
increasing size or because of increasing wettability.
The activated fractions of $CaCO_3$ particles after coating with oleic acid with vf of 16%
and 44% (coating temperatures of 70 and 80 ºC , respectively) were considerably
higher than that before coating, as shown in Fig. 4 (bottom panel). The increased
activated fractions resulted from both the increase in particle size (Fig. 3) and the
increase of the OA volume fraction of particles. At vf of 16% and 44%, the activated
fractions of the $CaCO_3$ particles after coating increased with the increase of SS and
reached complete activation. (Note, because the activation efficiency is 83%, the
activated fractions appear at values less than 100% at the points of full activation.)
For vf of 16% and 44% $SS_{crit}$ was determined by fitting a sigmoidal function to the
activated fraction as a function of SS. The particle dry diameter $D_P$ which is $D_{crit}$ in
these cases is given in Table 1. The hygoscopicity parameter $\kappa$ was determined from
$D_P$ ($D_{crit}$) and the corresponding $SS_{crit}$. The $\kappa$ values of the $CaCO_3$ particles coated
with vf of oleic acid of 16% and 44% were $0.0241 \pm 0.0006$ and $0.0649 \pm 0.0008$,
respectively. The respective $\kappa$ values for the $CaCO_3$ particles with a diameter of 101.9
nm without coating and after coating with oleic acid at 30-60 ºC (oleic acid vf $\leq 4.3\%$)
could not be determined by this method because these particles could not be fully
activated at the highest SS reachable by the CCN counter. Therefore we give $\kappa =$
$0.0028 \pm 0.0001$ for the uncoated $CaCO_3$ particles determined by scanning the size of
the poly-disperse $CaCO_3$ aerosol particles as described above (see Fig. 5).
So we conclude that for vf of oleic acid of 0.8-2.7% the CCN activity of $CaCO_3$
particles after coating is lower than that of uncoated $CaCO_3$ particles and decreases
with the fraction of oleic acid. The trend turns at a vf between 2.7 and 4.3%. CCN
activity was higher than that of the bare $CaCO_3$ particles at vf of oleic acid of 16%
(70°C) and 44% (80°C) with CCN activity $\kappa = 0.0241 \pm 0.0006$ and $\kappa = 0.0649 \pm 0.0008$,
respectively. The enhanced and reduced CCN activity of $CaCO_3$ particles coated with
oleic acid at 80 °C and 60 °C, respectively, was also evident from the CCN activity
measurement using *poly-disperse* aerosols (Fig. 5).
A possible explanation for our observation can be based on the amphiphilic character
of oleic acid, namely that one end of the oleic acid molecule is hydrophobic (the
hydrocarbon chain), while the other is hydrophilic (the carboxyl group).
We refer to $Ca(OH)(HCO_3)$ structures at the surface which offer polar surface sites to
bind the hydrophilic ends (the carboxyl groups) of the oleic acid molecules. The
hydrophobic ends of oleic acid molecules (the hydrocarbon chains) are then exposed
on the particle surface hence increase the hydrophobicity of the particle surface. Such
a formation of a hydrophobic layer should be occurring until all polar sites are
occupied or monolayer coverage is reached, maybe in form of a self-assembled layer.
This can hinder the uptake of water. Activation of $CaCO_3$ particles can be described
by the Kelvin term and a water absorption term, e.g. Frenkel Halsey Hill isotherm
(Sorjamaa and Laaksonen, 2007, Kumar et al., 2009). In terms of Kelvin/FHH theory
the hydrophobic OA coating will lower $A_{FHH}$ and/or likely increase $B_{FHH}$. The
formation of a monolayer of OA on black carbon particles with the polar groups
pointing outwards was postulated by Dalirian et al. (2017), which lead to increased
activation of the black carbon particles. Thus, they observed a similar effect of layer
formation, but with switched polarity.
Garland et al. (2008) suggested that OA at sub-monolayer coverage form
self-associated islands rather than uniformly covering the surfaces, and OA molecules
are oriented vertically, with polar heads facing to the surface. This is in support of our
working hypothesis: the formation of a hydrophobic surface film. We conclude that
all hygroscopic sides on the $CaCO_3$ surface are covered at OA vf somewhere between
2.7% and 4.3%, as here the trend turns and droplet activation starts to increase again.
This would place the monolayer coverage above 3% organic volume fraction.
According to the measurements and calculations of the length of oleic acid molecule,
the thickness of oleic acid sub-monolayer on solid surfaces, and the thickness of
deuterated oleic acid monolayers at the air-water interface (Garland et al., 2008; King
et at., 2009; Iwahashi et al., 2000), we estimate 2.3 nm as the likely thickness of oleic
acid monolayer on $CaCO_3$ particles, accordingly a monolayer would be achieved at
about 12-13% organic volume fraction. As a consequence the re-increase of
hygroscopicity starts at sub-monolayer coverage and we propose that a fraction of
oleic acid binds to already adsorbed oleic acid tail by tail such that carboxylic groups
are facing outwards.
For $CaCO_3$ particles coated with more than an OA monolayer (vf = 16% and 44% at
70 and 80 ºC coating temperatures), OA in the first layer should still combine with the
$CaCO_3$ surface, the heads pointing downwards. We suppose that now a portion of the
carboxyl groups of the oleic acid molecules, which are not in the first layer, will be
exposed to the particle surface, in analogy to the formation of lipid bilayers, e.g. in
cells, though the structure of this part of oleic acid is not known. The particle surface
then becomes more hydrophilic.
When carboxylic groups of OA are exposed at the surface, the interaction of water
with the OA layer becomes stronger, and the surface becomes wettable. In terms of
the Kelvin/FHH approach, the surface water interaction becomes stronger and $A_{FHH}$
increases and likely also the interaction between the higher water layers ($B_{FHH}$
decreases). From this point of view water adsorption by the "OA bilayer" should
become similar to thin malonic acid layers (compare next section). In addition, when
droplets form, oleic acid will transfer to the surface of the droplets and lower the
surface tension of the solution (the surface tension of oleic acid is 0.033 J m$^{-2}$, which
is much lower than that of pure water of 0.072 J m$^{-2}$). Thus, the activation of OA
coated particles is probably a complex interaction between formation of a specific
hydrophobic layers and more hydrophilic multilayers, surface tension effects and for
the largest coating amounts, simple size effects. As shown in Figure 7, $SS_{crit}$ for OA is
lower than for thin malonic acid coatings, probably because of the surface tension
effect, but higher than for thick MA coatings, because of the missing solute effect.
The CCN activity of all oleic coated particles is higher than the CCN activity of pure
oleic acid. Our CCN activity measurement showed that pure oleic acid particles up to
333 nm did not activate at 0.87% SS; this sets an upper limit for CCN activity of oleic
acid particles ($\kappa < 0.0005$), in agreement with Kumar et al. (2003) and Broekhuizen et
al. (2004). In liquid state oleic acid (OA) forms micelle like structures, the hydrophilic
ends (the carboxyl groups) of oleic acid molecules tend to combine together by
hydrogen bonds and the hydrophobic tails (the hydrocarbon chains) are exposed at the
outside (Iwahashi et al., 2000; Garland et al., 2008). The arrangement of oleic acid
molecules in pure oleic acid particles should be similar. Hydrophobic tails facing
outwards can explain the hydrophobicity of the particle surface and the hindrance of
the uptake of water, making the CCN activity of pure oleic acid particles very low.
For sub-monolayer coatings of OA of vf 0.8 - 2.7% the CCN activity seem to
approach that of pure OA. However, the arrangement of oleic acid molecules in these
thin coatings will be influenced by the $CaCO_3$ core with its polar, hydrophilic sides
differing from pure oleic acid particles and can thus be less hydrophobic.
Even at the largest coating with an organic volume fraction of 44%, the coating
thickness is about 10 nm, which corresponds to about only 4 monolayers of oleic acid
(assuming the thickness of oleic acid monolayer on $CaCO_3$ particles is about 2.3 nm).
And the arrangement of oleic acid molecules will still be likely influenced by the
$CaCO_3$ core. Water can probably adsorb at the carboxylic groups facing outward
("bilayer" type structure) and diffuse through the thin oleic acid coatings. It may form
an adsorbed water phase near the $CaCO_3$ surface. This could push the oleic acid out to
act as surfactant which lowers the Kelvin term. Such processes should also happen in
pure oleic acid particles. Because of the presence of $CaCO_3$ core the SS to achieve
this is lower than for pure OA.
The phenomenon described above is reported for the first time in the studies on the
CCN activity of multicomponent aerosols. This phenomenon also shows a limitation
of the otherwise very useful mixing rule (Petters and Kreidenweis, 2007) for
multicomponent aerosols with specific morphologies.
In Fig. 5 we additionally show the influence of water vapor on CCN activity of
$CaCO_3$ particles coated with oleic acid for the highest coating temperature (80 °C)
and thus largest oleic acid amount. Herein we determined $D_{crit}$ at different
supersaturations ($SS_{crit}$) for *poly-disperse* $CaCO_3$ aerosol particles (by SMCA). The
experiments were performed at RH 0.3% and at RH 3% at the coating temperature of
80 °C on cooling to room temperature the RH increased to 47%. The presence of
more water vapor (1500 Pa) in the coating process increased κ somewhat and
enhanced the CCN activity. This is of importance since RH will often larger than 0.3%
if coating appears in the atmosphere. This will be discussed further in context of
malonic acid coatings at enhanced water vapor.
**3.3 CCN activity of $CaCO_3$ particles with malonic acid coating**
For the study with malonic acid coatings, the $CaCO_3$ particles were also size selected
with a diameter of 101.8 nm. The size $D_P$ and chemical composition of $CaCO_3$ aerosol
particles are listed in Table 1 before and after coating with malonic acid (MA) at
temperatures in a range of 30-80 ºC. The mode diameter did not shift after coating in a
temperature range of 30-60 ºC, but it increased for coatings at 70 and 80 ºC with
increasing coating temperature. The size bin interpolated particle diameter $D_P$ of the
MA coated particles increased monotonically with the coating temperature. The
average of the interpolated diameter of bare $CaCO_3$ particles in the temperature range
30°C-80°C was $d_u$ = 101.9 nm.
Values of the malonic acid marker *m/z*42 per particle were significantly larger for
$CaCO_3$ particles after coating at 30-80 ºC and the MA mass increased from $3.3 \cdot 10^{-12}$
to $610 \cdot 10^{-12}$ µg per particle with the coating temperature (Table 1, Figure 3, bottom
panel). The organic volume fraction vf of malonic acid ($V_{MA}/(V_{MA}+V_{CaCO3})[\%]$) was
calculated as in the case of the oleic acid and ranged from 0.4 to 40%. As for oleic
acid the malonic acid experiments were repeated at least four times and the
reproducibility and stability were good (see standard deviations in Table 1).
The activated fractions at different SS for 101.9 nm $CaCO_3$ particles before and after
coating with malonic acid at 30-80 ºC are shown in Fig. 6. $SS_{crit}$ was determined by
fitting a sigmoidal function to the data and the $\kappa$ value was calculated from the
$D_P(D_{crit})$ and the corresponding $SS_{crit}$. The results are listed in Table 1. In this
procedure we had to neglect the contribution of double charged particles as the step in
the CN/CCN vs SS data in Fig.6 is not sufficiently expressed to separate a plateau for
multiply charged particles (e.g. Sullivan et al., 2009). The exception is the MA
coating with vf = 0.4%. For this case we compared a sigmoidal fitting both from the
beginning (the first point) and from the multiply-charged plateau (the third point) to
the "completely-activated plateau" (Figure S3, supplement). We yield $SS_{crit}$ =
0.887±0.005% for fitting from the beginning and $SS_{crit}$ = 0.900±0.013% for fitting
from the multiply-charged plateau, a difference of 0.013%. The underestimate in $SS_{crit}$
is the largest (0.013%) when the MA mass is the smallest (vf = 0.4%) and the
underestimate will be reduced with increasing vf of MA. At the largest two MA vf it
can be neglected. We have to concede a systematic error in $SS_{crit}$, but it is distinctively
less than 0.02%.
The $\kappa$ values of the $CaCO_3$ particles after coating with malonic acid at 30-80 ºC were
higher than the $\kappa$ value of the uncoated $CaCO_3$ particles ($\kappa = 0.0028 \pm 0.0001$), and
increased with the increasing coating MA mass per particle and increasing MA vf.
The CCN activity of the $CaCO_3$ particles increased monotonically after coating with
increasing malonic acid mass. This result differs from that of oleic acid which is not
surprising since malonic acid is easily soluble in water.
The $\kappa$ value for the $CaCO_3$ particles after coating with a mass of malonic acid as small
as $3.3 \cdot 10^{-12}$  µg per particle and vf of MA of only 0.4% was $0.0123 \pm 0.0005$ thus
considerably larger than the $\kappa$ value for the uncoated $CaCO_3$ particles ($\kappa = 0.0028 \pm$
0.0001). This suggests already that a small amount of malonic acid can significantly
enhance the CCN activity of $CaCO_3$ particles. Such phenomenon, that traces of water
soluble substances can strongly affect droplet activation has been reported before
(Bilde and Svenningsson, 2004).
We applied Koehler theory to $CaCO_3$ particles coated with malonic acid assuming
that the malonic acid coating will fully dissolve in water when droplets form (see
supplement eq. (S1-S3). Increasing MA solute decreases the activity of water in
solution, and lowers the critical supersaturation $SS_{crit}$ for droplet activation.
The resulting Koehler curves, i.e. equilibrium supersaturation (SS) over the solution
droplet as a function of the wet diameter $D_w$, are shown in Figure S2. Therein the
maximum of each SS curve is the critical supersaturation (theory $SS_{crit}$). In Table S1
and Figure 7 we compare the $SS_{crit}$ predicted by the Köhler approach (red) with the
observed $SS_{crit}$ (black). Koehler theory overpredicts $SS_{crit}$ for thin coatings
substantially, meaning it underestimates the hygroscopicity of the thinly coated
particles. But with increasing coating Koehler theory approaches the observed $SS_{crit}$
and $SS_{crit}$ for a particle of 121 nm diameter composed of pure malonic acid is the
limiting case (red circle).
From the Koehler results we derived the water content of the particles at $SS_{crit}$ and we
calculated molality and mass fraction of the solute in the solution at the point of
activation. The molality at minimum and maximum malonic acid load of $3.3 \cdot 10^{-12}$
μg/particle and $610 \cdot 10^{-12}$ μg/particle were 0.006 mol kg$^{-1}$ and 0.0015 mol kg$^{-1}$,
respectively. We used these values in the AIOMFAC model (Zuend etal. 2011) to
calculate the deviation from ideality for the solution at point of activation for a flat
solution. The both solutions are highly non-ideal with respect to the MA ($a_x = 0.4$),
wherein MA was treated as solute with reference state infinite dilution (mole fraction
$x_{solute} \rightarrow 0$). However this did not affect much the activity coefficient of water, which
is essentially 1, water treated as solvent with reference state pure liquid (mole fraction
$x_{water} = 1$). Moreover, in this concentration range, the surface tension of aqueous
malonic acid solutions is about 0.070 J m$^{-2}$, thus the nearly same as for water (Table
S2 in the supplement). One should expect that Koehler theory would predict SS quite
well under such conditions.
To bring Köhler theory in agreement with the observation for the thinnest coating,
*more* solute entities would be required. Thus, disagreement cannot be caused by an ill
determined van't Hoff factor as we used already maximum $v = 3$ and reducing $v$ will
increase the deviation. Note, that recent observations point to the importance of the
surface effect by organic surface films over the solute effect for water soluble
inorganics in presence of organics, including malonic acid (Ruehl et al., 2016). A
lower surface tension will bring Koehler prediction and observation punctually in
better agreement and still allow for smaller van't Hoff factors (Varga et al., 2007). As
an example, a surface tension of 55% of $\sigma_w$ and a van't Hoff factor of one will bring
SS$_{crit}$ predicted by Koehler theory and observation in agreement for the thinnest
coating. However, a surface tension 55% of $\sigma_w$ will cause disagreement for the
thickest coating, because the solute term gains in importance. Probably, the findings
for the mixed solutions of malonic acid and water soluble ammonium sulfate are not
directly transferable to our systems with insoluble inorganic core, where we expect
dilute aqueous solutions of 0.006 mol/kg of malonic acid at the activation point. At
such concentrations malonic acid does not reduce $\sigma_w$, moreover in the study of Ruehl
et al. (2016) malonic acid was one of the more Koehler $\kappa$ behaving organics.
In Figure 7 we show the prediction of SS$_{crit}$=1.52% for activating CaCO$_3$ by the
Kelvin/FHH theory with the $CaCO_3$ parameters taken from Kumar et al. (2009). $SS_{crit}$
for our bare $CaCO_3$ particles is 1.49% and the lower $SS_{crit}$ should be due to a more
adsorptive surface, e.g. the presence of $Ca(OH)(HCO_3)$ structures. According to
classical Koehler theory the equivalent of $8.5 \cdot 10^{-20}$ moles of dissolvable entities
would be needed to explain a $\kappa$ of 0.0028 and $SS_{crit}$ of 1.49%, which is only about ¼
of the moles MA in the thinnest MA coating. Therefore, whatever makes our $CaCO_3$
particles wettable is not sufficient to explain the low $SS_{crit}$ of 0.9 % at the thinnest MA
coating in terms of Koehler theory.
We estimate monolayer coverage by MA at 2-3% vf; this would be achieved in
between MA mass loads of $13 \cdot 10^{-12}$ - $38 \cdot 10^{-12}$ μg per particle. Thus a sub-monolayer
coating of $3.3 \cdot 10^{-12}$ μg MA per particle caused a drop of $SS_{crit}$ from 1.49 to 0.9 and
increased $\kappa$ from 0.0028 to 0.012. Therefore we conclude that $CaCO_3$/MA coatings
show a non-Koehler behavior at thin coatings, but approach Koehler behavior with
increasing MA load.
This means there must be specific interactions between MA and the $CaCO_3$ surface
which eases water adsorption and CCN activation. We refer to the $Ca(OH)(HCO_3)$
structures that likely exist on the particle surface. When $CaCO_3$ particles are coated by
malonic acid (or oleic acid) the hydrophilic sides can serve as polar surface active
sites for accommodation of the acids. In case of MA there is no long hydrophobic
organic chain, but a second carboxylic group which still could support the adsorption
of water films.
In terms of Kelvin/FHH theory one could explain the observed low $SS_{crit}$ for thin MA
coatings by net stronger interaction with water (higher $A_{FHH}$) and/or stronger
interaction between the adsorbed water layers (lower $B_{FHH}$) compared to bare $CaCO_3$.
If coatings become thicker the Koehler solute effect starts increasingly to contribute
and eventually controls the CCN activation. Our data are not sufficient to determine
$A_{FHH}$ and $B_{FHH}$. (The only system in the literature which comes close - in a far sense -
is CaOxalate Monohydrate, with $A_{FHH} = 0.57$ of and $B_{FHH} = 0.88$ (Kumar et al., 2009).
Plugin in these FHH parameters will lead to $SS_{crit} = 0.53\%$ commensurable with our
observed value of 0.56% for $13 \cdot 10^{-12}$ μg MA coating which represents an organic
volume fraction of 1.5%, thus is close to monolayer.)
The $D_{crit}$ at different supersaturations ($SS_{crit}$) for *poly-disperse* $CaCO_3$ aerosol
particles before and after coating with malonic acid are shown in Fig. 8. Our
observation of $κ = 0.25 \pm 0.04$ for pure malonic is consistent with the κ derived from
the data of Kumar et al. (2003) ($κ = 0.20 - 0.25$) and Prenni et al. (2001) ($κ = 0.24$), but
significantly lower than the κ derived from the data of Giebl et al. (2002) ($κ =$
0.41-1.04). The behavior of poly-disperse coated aerosol was similar to the result
obtained from the monodisperse $CaCO_3$ aerosol particles.
In Fig. 8 we added results for coating in presence of enhanced water vapor (1500 Pa)
and aerosols generated by spraying mixtures of malonic acid and $CaCO_3$. At the
coating temperature of 60 ºC, when the RH increased from 0.7% to 7% and eventually
to 47% at room temperature, the CCN activity of the coated $CaCO_3$ particles
increased substantially (compare "dry" (green triangles) and "wet" (lilac triangles) in
Fig. 8). The effect is more distinct than for the oleic acid coating shown in Fig. 5, and
$\kappa$ increases by about an order of magnitude. At a wet conditions, the reaction between
$CaCO_3$ and malonic acid maybe more efficient and formation of calcium malonate
will reduce $d_u$, i.e. the diameter of the insoluble core and according to eq, (S1) this
may be the reason for the higher CCN activity at the higher RH. The hypothesis of
malonate formation is supported by the CCN activity of "calcium malonate" aerosols,
generated by spraying solutions containing $CaCO_3$ and malonic acid with molar ratios
of about 1:1 and 3:1. Here the CCN activity is similar to that arising in the coating
process in presence of water vapor. The change of the Ca/malonate ratio from 3:1 to
1:1 had no large effects. But taking the data of pure malonic acid particles also into
account there is a trend to lower $\kappa$ with increasing Ca in the initial solution.
The increasing of residence time (by 23.7 s) had no significant impact on CCN
activity for both oleic acid coating and malonic acid coating at both dry and enhanced
water vapor conditions, probably because the coating process was already completed
in the coating device and no further reactions occurred in the flow tube.
Our findings may be important for aging processes of mineral particles in the
atmosphere. The dependence of CCN activity of the coated particles on RH during the
coating process will help to enhance the increase of the CCN activity by the coating
process as water will be abundant in many instances. The effect probably will be
relatively small for oleic acid and similar organics, which are hardly water soluble,
but strong for malonic acid and similar organic acids, which are highly water soluble.

## 4 Conclusions

The CCN activity of $CaCO_3$ particles with oleic acid and malonic acid coatings was
investigated in this study. The results show that oleic acid coating and malonic acid
coating have different impacts on the CCN activity of $CaCO_3$ particles. This can be
attributed to the amphiphilic property of oleic acid in contrast to the high water
solubility of malonic acid. Small amounts of oleic acid coating (vf $\leq$ 4.3%) decreased
the CCN activity of the $CaCO_3$ particles, while more oleic acid coating (vf $\geq$16%)
increased it. This phenomenon was reported here for the first time, and attributed to
stepwise passivating the active sites of $CaCO_3$ by oleic acid. Once all active sites are
occupied we suggest the formation of a lipid like bilayer with the carboxylic groups
facing outwards.
On the other hand, malonic acid coating (0.4-40%) increased the CCN activity of
$CaCO_3$ particles regardless of the amount of the coating. The CCN activity of $CaCO_3$
particles with malonic acid coating increased with the amount of the coating. Even a
small amount of malonic acid coating (vf = 0.4%) significantly enhanced the CCN
activity of $CaCO_3$ particles from $\kappa = 0.0028 \pm 0.0001$ to $\kappa = 0.0123 \pm 0.0005$.
Increasing the relative humidity during the coating increased the CCN activity of the
$CaCO_3$ particles with malonic acid coating, probably because more $CaCO_3$ reacted
with malonic acid to soluble CaMalonate. This process will help to increase the CCN
activity.
Although malonic acid is well soluble in water, $SS_{crit}$ for MA coated particles was
overpredicted by Köhler theory. Our results indicate that thin MA coatings provide a
wettable particle surface, which favors adsorption of water. For thicker coatings the
coated particles approached Köhler behavior, because of increasing importance of the
solute effect.
Mineral aerosol is one of the most abundant components of the atmospheric aerosol,
but its low water solubility limits its CCN activity. This study showed that
water-soluble organic acid coating might significantly enhance the CCN activity of
mineral aerosol particles. This could lead to mineral aerosol playing a more important
role in cloud formation.

**Acknowledgments**
This study was supported by Forschungszentrum Jülich, the National Natural Science
Foundation Committee of China (41421064, 21190051, 41121004), and the China
Scholarship Council.

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

**Table 1.** Mode diameters, chemical compositions, and κ values of $CaCO_3$ aerosol particles (size selected by DMA at 101.8 nm) before (uncoated) and after coating with oleic (OA) or malonic acid (MA) at 30-80 ºC.

| | $D_P$ [nm] | Organic mass per particle $[10^{-12}\mu g]$ | Mole organics per particle $[10^{-20}$ mole] | Org. volume fraction [%] | κ |
|---|---|---|---|---|---|
| **Oleic acid** | | | | | |
| **Uncoated $CaCO_3$ 30-80 ºC** | 101.9 | (backgr. $m/z$41: 2.7±0.9) | 0 | 0.0 | 0.0028±0.0001 |
| **$CaCO_3$+Oleic acid 30 ºC** | 102.1 | 3.7±1.9 | 1.3 | 0.8 | |
| **$CaCO_3$+Oleic acid 40 ºC** | 102.5 | 7.0±2.8 | 2.5 | 1.4 | |
| **$CaCO_3$+Oleic acid 50 ºC** | 103.7 | 14±3.7 | 5.1 | 2.7 | |
| **$CaCO_3$+Oleic acid 60 ºC** | 104.9 | 23±1.2 | 8.3 | 4.3 | |
| **$CaCO_3$+Oleic acid 70 ºC** | 109.2 | 96±3.7 | 34 | 16 | 0.0241±0.0006 |
| **$CaCO_3$+Oleic acid 80 ºC** | 123.7 | 390±14 | 140 | 44 | 0.0649±0.0008 |
| | | | | | |
| **Malonic acid** | | | | | |
| **Uncoated $CaCO_3$ 30-80 ºC** | 101.9 | (backgr. $m/z$42: 1.4±0.4) | 0 | 0.0 | 0.0028±0.0001 |
| **$CaCO_3$+Malonic acid 30 ºC** | 102.0 | 3.3±0.3 | 3.2 | 0.4 | 0.0123±0.0005 |
| **$CaCO_3$+Malonic acid 40 ºC** | 102.1 | 6.8±1.2 | 6.5 | 0.8 | 0.0231±0.0008 |
| **$CaCO_3$+Malonic acid 50 ºC** | 102.2 | 13±1.8 | 13 | 1.5 | 0.0380±0.0012 |
| **$CaCO_3$+Malonic acid 60 ºC** | 102.7 | 38±1.6 | 36 | 4.1 | 0.1063±0.0023 |
| **$CaCO_3$+Malonic acid 70 ºC** | 107.8 | 160±8.1 | 160 | 15 | 0.1907±0.0031 |
| **$CaCO_3$+Malonic acid 80 ºC** | 121.0 | 610±24 | 590 | 40 | 0.3126±0.0062 |

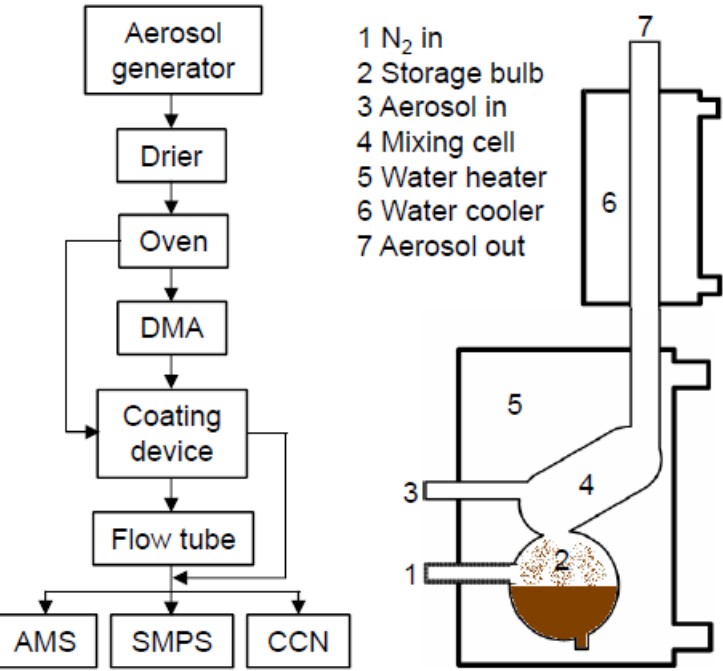

**Figure 1.** Schematics of the experimental set up (left side). CaCO$_3$ aerosol is generated by spray-drying of saturated Ca(HCO$_3$)$_2$ solutions and tempering the aerosol passing through an oven at 300 °C. The *poly-disperse* CaCO$_3$ aerosol is either led directly to the coating device (right side, after Roselli, 2006) or led to a differential mobility analyzer (DMA) for size selection first. Optional, a flow tube can be switched into the pass to enhance the reaction time of the coated particles. The stream of coated particles is finally split to the analytical instruments, namely aerosol mass spectrometer (AMS), scanning mobility particle sizer (SMPS) and CCN counter.

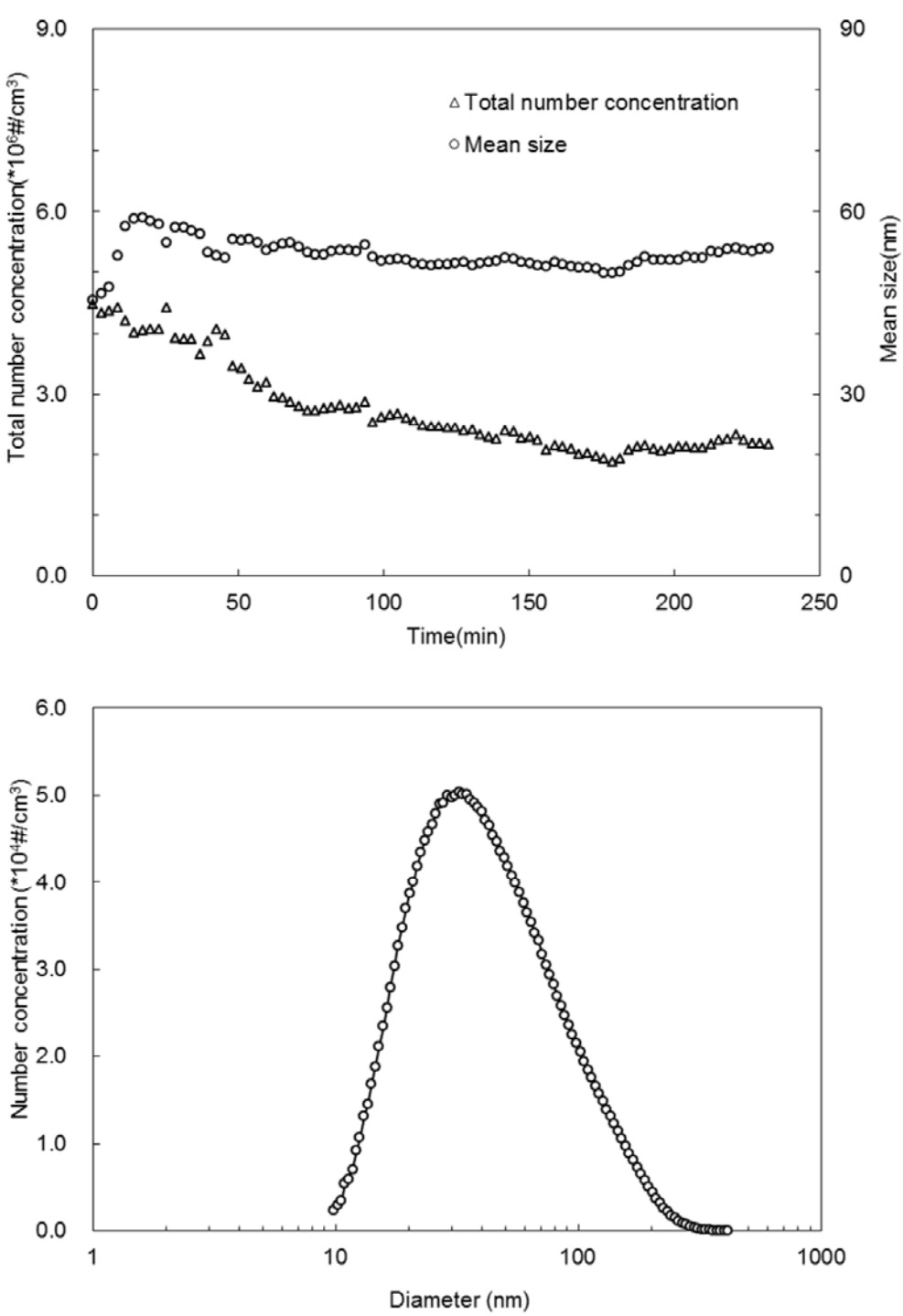

**Figure 2.** Total number concentration and mean diameter of $CaCO_3$ aerosol particles generated as a function of the spraying time (upper panel). Typical size distribution of the $CaCO_3$ aerosol after 70 min spraying (lower panel).

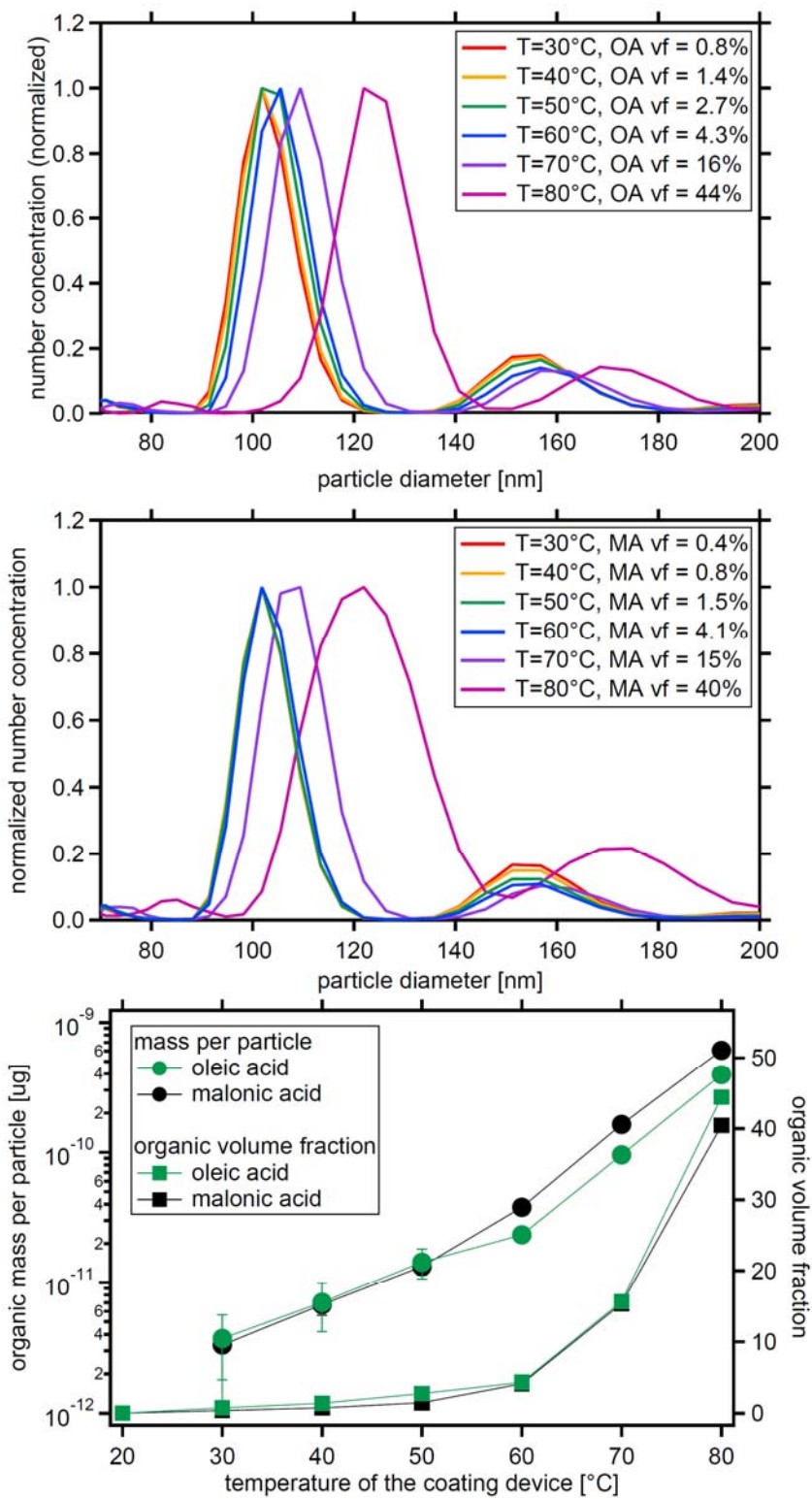

**Figure 3.** Size distribution of monodisperse CaCO₃ aerosol particles after coating with oleic acid (top panel) or malonic acid (middle panel). Coating amount and organic volume fraction for oleic and malonic acid as a function of the coating temperature for the same experiments (bottom panel, data in Table 1).

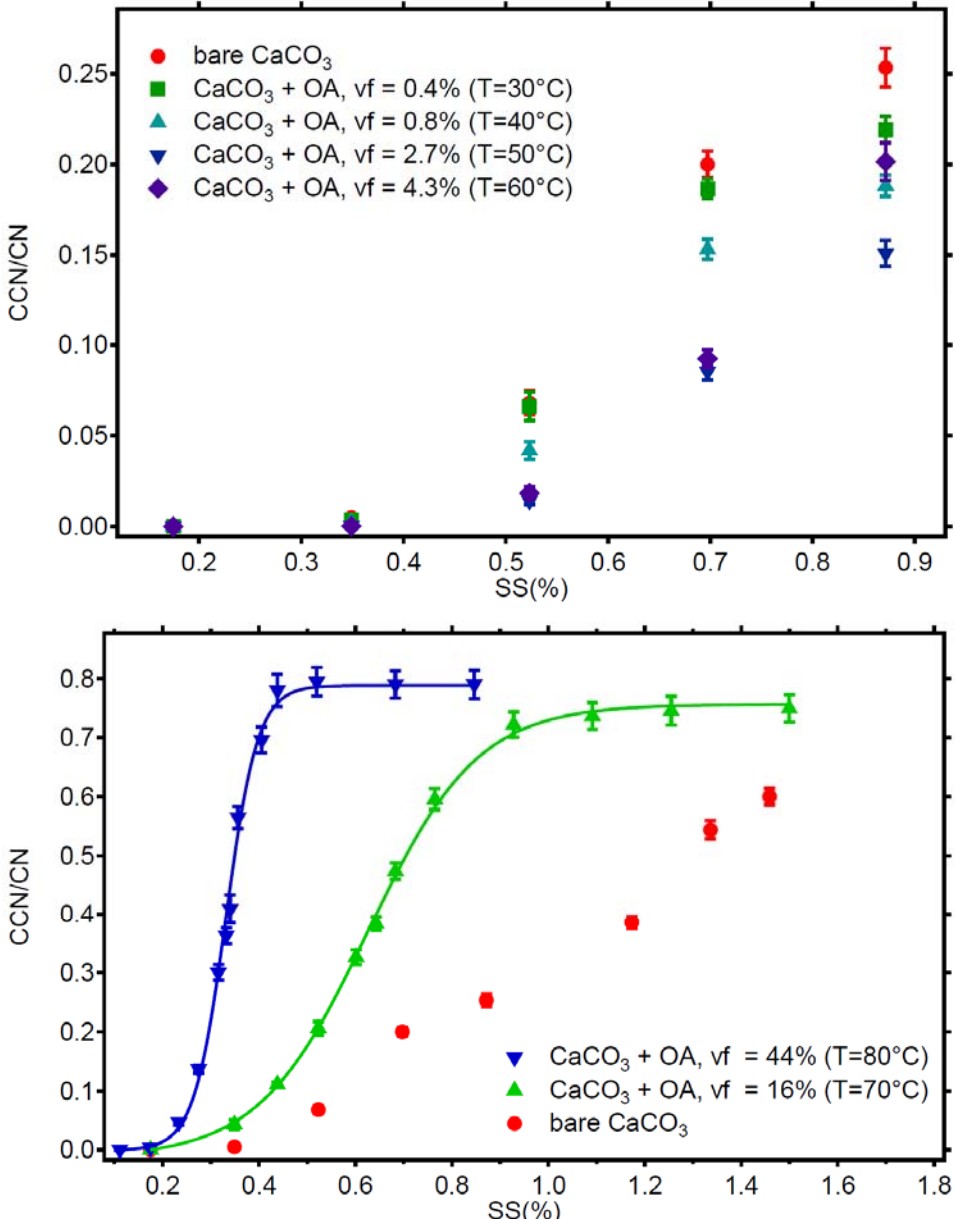

**Figure 4.** Activated fractions (CCN/CN) of *monodisperse* CaCO$_3$ aerosol particles (diameter $d_u$ = 101.9 nm) at different supersaturations before and after oleic acid coating. With increasing coating temperatures of 30-50 ºC the activated fraction decreases despite the increase of organic vf from 0.8% - 2.7%. At vf = 4.3% at 60 °C this trend turns. Considering the increased particle diameter at 30-60 ºC and the reduced activated fraction simultaneously, the CCN activity of the coated CaCO$_3$ particles at 30-60 ºC was lower than that of the uncoated CaCO$_3$ particles (top panel). At coating vf of 16% and 44% (T= 70-80 ºC) the activated fractions, thus CCN activities, are higher than for bare CaCO$_3$ and increase with coating vf. In these two cases all particles are activated at the highest SS and SS$_{crit}$ and $\kappa$ can be determined from the turning point of the sigmoidal fit (bottom panel, compare Table 1).

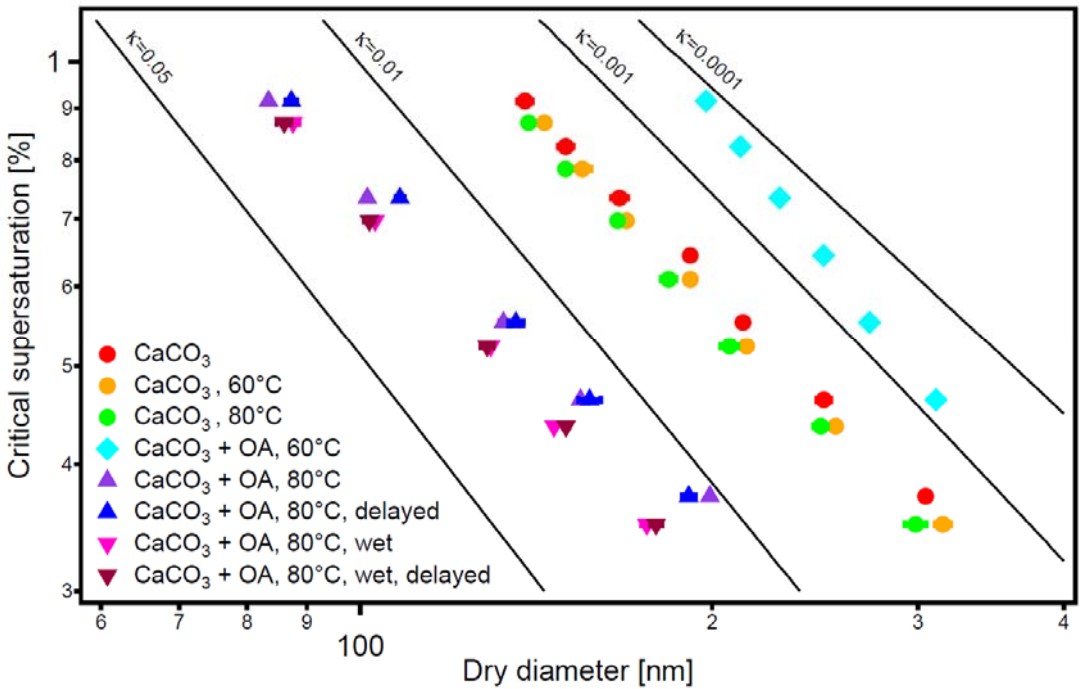

**Figure 5.** Critical dry diameters at different supersaturations (SS) of *poly-disperse* CaCO₃ aerosol before (circles) and after oleic acid coating. Experiments were performed at 60°C (turquoise diamond) and 80°C (triangles) coating temperatures. The flow tube experiments at 80° C were performed (indicated by 'delayed') at dry conditions (normal, blue tringles) and at enhanced water vapor ('wet', brown triangles). The effect of the temperature in the coating device on the CaCO₃ core is negligible (red, green, and orange circles). As for the monodisperse case in Figure 4, at 60° coating temperature the particles are less CCN active than bare CaCO₃ while at 80°C the coated particle more CCN active. The presence of water vapor (1500 Pa) in the coating process enhances CCN activity. The increasing of residence time (23.7 s) of the coated aerosol in the flow tube had no significant effect on CCN activity in both experiments.

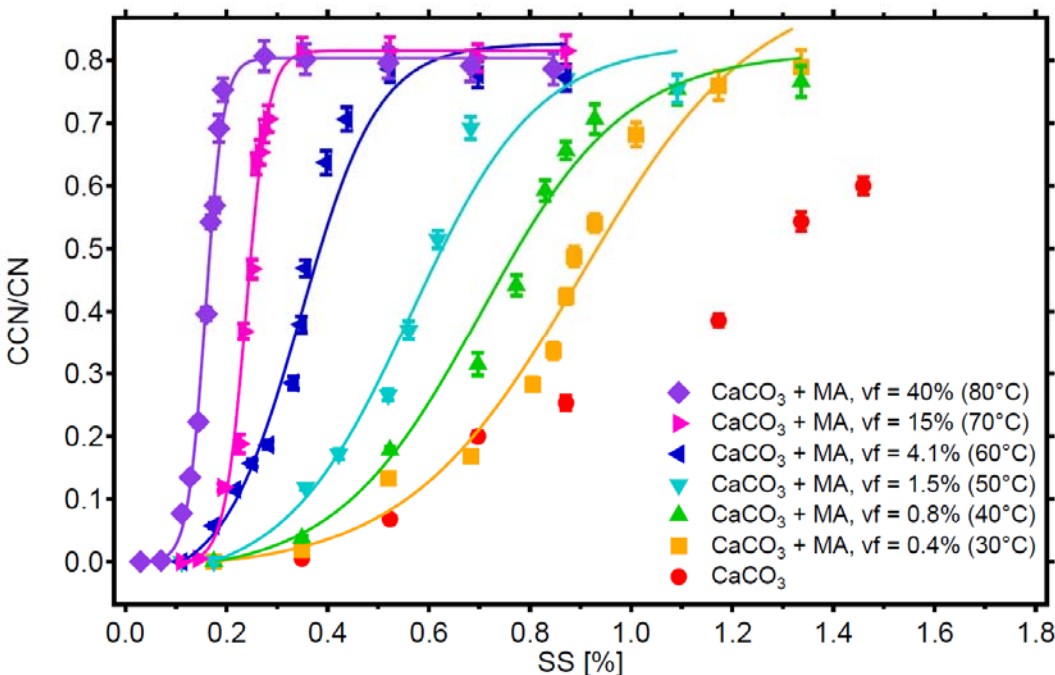

**Figure 6.** Activated fractions (CCN/CN) of *monodisperse* CaCO$_3$ aerosol particles (with CaCO$_3$ core, d$_u$ = 101.9 nm) at different supersaturations before (red circles) and after malonic acid coating. With increasing coating, i.e. MA volume fraction vf the activated fraction, thus CCN activity, increase compared to bare CaCO$_3$ particles. All coated particles can be activated at sufficiently high SS and SS$_{crit}$ and $\kappa$ was determined (see Table 1).

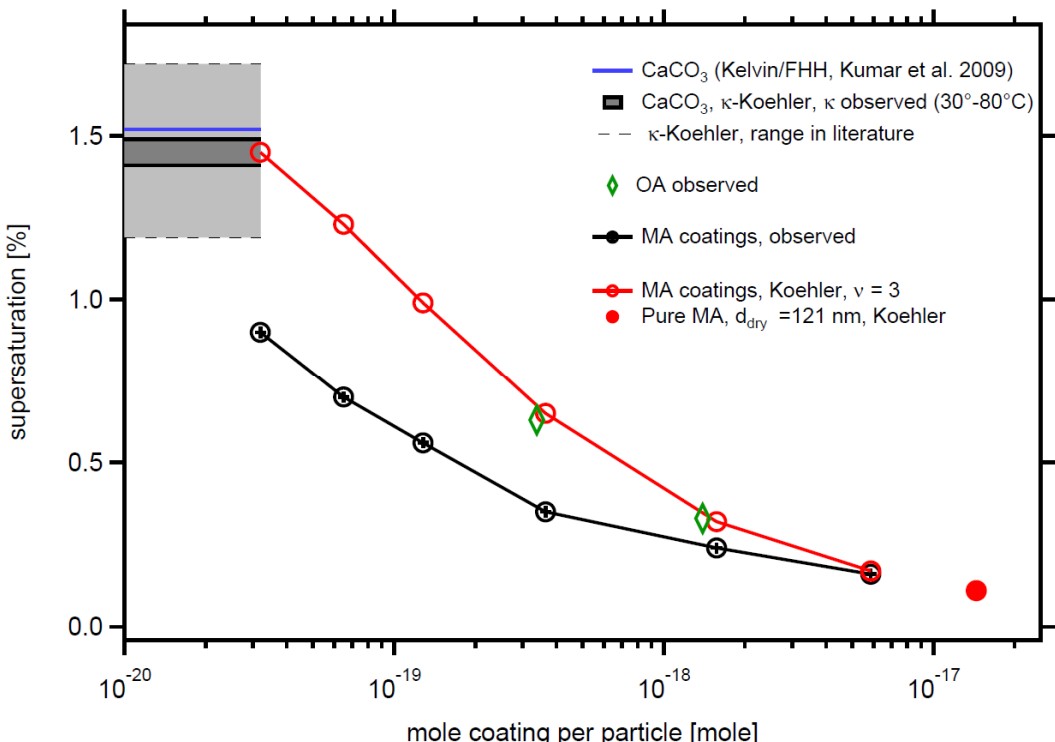

**Figure 7.** Comparison of $SS_{crit}$ predicted by Koehler theory with observations. Koehler theory for aqueous MA solutions assuming full dissociation ($\nu=3$) overpredicts $SS_{crit}$ (red circles) compared to the observation (black circles). With increasing coating amount Koehler theory approaches the observation, with the limiting $SS_{crit}$ for 121.0 nm particles made of pure malonic acid. For comparison we show also observed $SS_{crit}$ for the two thickest OA coatings (green diamonds). The horizontal lines indicate $SS_{crit}$ of the bare $CaCO_3$ particles as calculated from our observed $\kappa$ (black) and predicted by Kelvin/FFH theory (blue). Light grey area between the thin dashed grey lines shows the range of $SS_{crit}$ for 101.9 nm particles calculated from the range of $\kappa$ in literature for wet generated $CaCO_3$ particles (compare Tang et al. 2016).

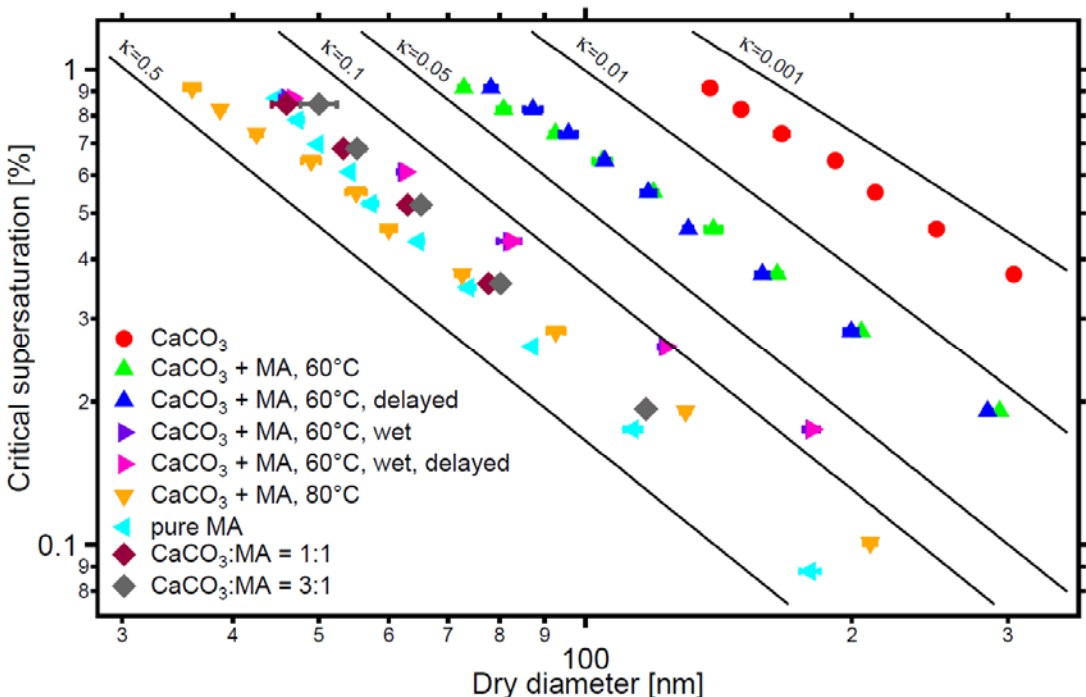

**Figure 8.** Critical dry diameters at different supersaturations (SS) of *poly-disperse* CaCO₃ aerosol before (circles) and after malonic acid coating (triangles). Experiments were performed at 60°C and 80°C coating temperatures. The results are similar to the monodisperse case in Figure 6. Critical dry diameters as a function of SS are also shown for malonic acid particles and particles that were generated by spraying mixed solution with molar ratios of CaCO₃/malonic acid of 1:1 and 3:1. The CCN activity decreases with increasing CaCO₃ content. The flow tube experiments at 60 °C were performed (indicated by 'delayed') at dry condition (blue tringles) and in presence of 1500 Pa water vapor (magenta triangles). The presence of water in the coating process substantially enhanced κ and CCN activity. The increasing of residence time (23.7 s) of the coated aerosol in the flow tube had no significant effect on CCN activity in both experiments.