# Peer review of "Mingjin Wang1,2, Tong Zhu1\*, Defeng Zhao2, Florian Rubach2,3, Andreas"

_Atmospheric Chemistry and Physics, 2017_

## Short Comment (SC1) · 20 Nov 2017

May I bring authors' attention to a review paper (Tang et al., 2016) we published very recently? In this review paper we have provided a comprehensive review of CCN activity of fresh and aged CaCO3 particles (as well as other minerals).

References: Tang, M. J., Cziczo, D. J., and Grassian, V. H.: Interactions of Water with Mineral Dust Aerosol: Water Adsorption, Hygroscopicity, Cloud Condensation and Ice Nucleation, Chem. Rev., 116, 4205–4259, 2016.
* * *

---

## Referee Comment (RC1) · Anonymous Referee #2 · 18 Dec 2017

Overall assessment:

In this paper the effect of coating of oleic acid (low solubility in water) and malonic acid (water soluble) on the activation of particles to cloud drops is investigated. The CCNC experiments seem to be carried out according to present practice and the coating is done with great care.

It is thus disappointing that data evaluation does not meet the same standards. I have mainly identified two areas in which I would have liked to see a deeper analysis:

1) A full Köhler theory treatment based on the chemical composition of the particles would have been useful. A theoretical consideration of the expected changes in kappa values due to the reactions suggested and bilayer of oleic acid would improve the

paper.

2) Focusing on the coating device temperature in the results section is a bit confusing and I am not convinced about the reproducibility of the experiment, given that the experimental cases are identified mainly by the temperature of the coating device. Focusing more on presenting the mass of organics coated on the particles would improve the paper significantly. Does the coating thickness depend on the original particle size? Can this give more information towards understanding the observations?

Also, there are several cases where the text is not very clear. I have indicated some of them below.

For these reasons I recommend that the paper is not published in its present state, but that it is considered again after major revision.

Detail comments:

In the motivation of the chosen substances: is oleic acid really a good choice for a surface-active compound, considering its low solubility?

On line 179, the residence time in the flow tube is given. However in a laminar flow, the residence time varies with the position, i.e. whether the air passes at the centre or near the walls. Is this effect considered and may it influence the results?

Lines 180 to 185: At first reading it sounds as if the relative humidities given are at 80C. However, in the case of 47% RH, the air would then be strongly supersaturated at 25C. Please clarify the temperature at which the RH is given.

Also, if 47% RH refers to 25C, the RH at 80C will still be very low, just a few percent. How does this influence the experiments?

In lines 228 to 231 proportionalities between AMS signals and organic particle mass and volume are described. Are they observations or assumptions?

Also, determination of the amount of organic coating: If I read the table 1 correctly, and

make some calculation, the change in diameter and the change in organic mass, do not agree. Is this correct? Why is that?

Line 234 to 235. Have you tried to account for mode shift within the bin, by fitting a function to the distribution?

Line 293:You say " and our kappa value is somewhat higher but still in this range". Isn't it just in the range?

Line 298: You say " The increased kappa value of 0.0008". Shouldn't it be "The increase in kappa value of 0.0008"?

Line 323: What do you mean by a "significant amount" of coating? Is a coating of less than 2 nm insignificant? In all aspects?

Line 323-325: Unclear expression. Please specify that the sizing according to mobility diameter is referred to (if that is the case) and specify the temperature limit, instead of saying " a certain value".

Line 371: Why does the activated fraction level off at 83%? Did you make an intercalibration between numbers in the CCNC and CPC for other compounds?

In section 3.2 there is a lot of repetition, I think. Please see if the text can be made more efficient.

Line 413, the sentence starting with "This suggestion is supported by very low CCN activity of pure oleic acaid.....". Is this based on some calculations? Please describe.

Line 421 to 423: But the low surface tension does not seem to help the pure oleic acid particles to activate, according to your measurements as well as Kumar et al. (2003) and Broekhuizen et al. (2004). Also, the first "of" on line 422 should be removed.

Figure 6: The fitted curves for 30-60C do not seem to follow the data very well. Why? Have you accounted for doubly charged particles? In my opinion there is a tendency to a two step-function, the first step reaching to CCN/CN of 0.15-0.2.

[Figure]

---

## Short Comment (SC2) · 2 Jan 2018

Thanks for your information! It is a very good and impressive review paper!

---

## Referee Comment (RC2) · Anonymous Referee #1 · 24 Jan 2018

The CCN activity of CaCO3 particles with oleic acid and malonic acid coatings was investigated in this study. The results show that oleic acid coating and malonic acid coating have different impacts on the CCN activity of CaCO3 particles. This can be attributed to the amphiphilic property of oleic acid in contrast to the high water solubility of malonic acid.

Indeed, malonic acid behaved as expected, with small amounts drastically impacting the CCN properties of CaCO3 particles, with some impact of ageing processes involving water.

The behavior of oleic acid is more surprising, first suppressing or reducing the CCN properties for very thin coatings, and then enhancing them at multilayer coverages. This is tentatively explain by a kind of a bilayer being formed on top of the CaCO3

particles, first exposing the hydrophobic tail to water (reducing effects) and then the carboxylic end groups (enhancing effects). This is a quite interesting effect, showing non-linear impacts of the coatings on the CCN properties.

This topic is therefore fully suitable for publication in ACP, and I would recommend its publication once the authors have dealt with the following points. I also underline that I'm not an expert in CCN properties measurements.

- Overall the paper is nicely illustrated but it contains some repetitive text blocs, especially in the result section, that could be avoided in order to ease the reading of this manuscript (and reinforcing also its content).

- While the argument of a bilayer of oleic acid does make sense, I'm still puzzled by the fact that the CCN properties are apparently higher for a coated particle compared to a pure oleic acid particle. I would have simply assumed that once thick enough, the water probing the surface does not see the core $CaCO_3$ particle (over the time scale of these experiments). In this situation, the pure oleic acid particle would exhibit a kind of upper limit for water adsorption and droplet activation. Maybe the authors could comment more on that, and maybe add the pure oleic acid data on their figures (this would ease the comparison with both systems).

- If my reading is correct, the $CaCO_3$ particles with a thick coatings do exhibit better CCN properties than the pure oleic acid particles (if my reading is incorrect, this would highlight that an in-depth editing would be beneficial for the reader). How can you explain such a fact? Is the $CaCO_3$ surface leading to some kind of ordering of the adsorbed organic acid (which might not be observed in the pure homogeneous organic particle) leading effectively to the above mentioned bilayer structure?

- Is the temperature in the coating device leading to some kind of ordering (for instance, by increase surface mobility before an ordering when cooling down)? Also, oleic acid has only a moderate thermal stability as it decomposes at higher temperature (typically at temperature at a factor 2 higher than those used here), potentially via an epoxide

pathway in presence of air (and enhanced by light or metallic traces). Could traces of oxidized products nevertheless affect the composition of the coating at 80°C already? If so, then this effect should increase with time. Did the authors observed any variation with time of kappa at 80°C? Did the authors tried to have thicker coatings at lower temperature by changing the gas flow conditions in their coating device?

- Line 238. This sentence is unclear.

- Line 249: where a polydisperse

- Line 296: remained at

---

## Author Comment (AC1) · 27 Mar 2018

**Cloud Condensation Nuclei Activity of CaCO₃ Particles with Oleic Acid and Malonic Acid Coatings**

**Mingjin Wang**[1,2], **Tong Zhu**[1*], **Defeng Zhao**[2], **Florian Rubach**[2,3], **Andreas Wahner**[2], **Astrid Kiendler-Scharr**[2], **and Thomas F. Mentel**[2*]

*In this paper the effect of coating of oleic acid (low solubility in water) and malonic acid (water soluble) on the activation of particles to cloud drops is investigated. The CCNC experiments seem to be carried out according to present practice and the coating is done with great care.*

*It is thus disappointing that data evaluation does not meet the same standards. I have mainly identified two areas in which I would have liked to see a deeper analysis:*

We thank the reviewer for the positive and critical remarks. We will address the critical remarks in the following.

**Tables and Figures for illustration of the responses are given in an appended document.**

*C1. A full Koehler theory treatment based on the chemical composition of the particles would have been useful. A theoretical consideration of the expected changes in kappa values due to the reactions suggested and bilayer of oleic acid would improve the paper.*

**Response:**

The advantage of using kappa approach (Petters and Kreidenweis, 2007) is that all differences in the (underdetermined) system are mapped to one parameter (i.e. the ratio of the partial molar volumes of water and the solute(s)) and thus CCN activation of differently composed particles becomes comparable. We would like to note, that in Figure 5 and 7 we show the observations in the domain critical supersaturation vs dry diameter relative to fixed kappa lines. For ideal solutions at activation the data in these plots should "parallel" the kappa lines, thus represent one kappa. Deviation from kappa lines indicates non-ideality and other limitations of the kappa approach.

The Koehler theory is described in details by Koehler (1936). For applying Koehler theory to $CaCO_3$ particles coated with malonic acid at dry condition (RH < 0.7%), we assume that the malonic acid coating will fully dissolve in water when droplets formed. This decreases the activity of water in solution, and thus increases the CCN activity. We assume that at the critical supersaturation ($SS_{crit}$) the solution is ideal, i.e. water activity coefficient is close to 1, and the partial molar volume of water equals the molar volume of pure water $M_w/\rho_w$. Since the solubility of $CaCO_3$ in water is very low (0.00058 g/100 g water, at 298 K), while the solubility of malonic acid (MA) is quite high (62 g/100 g water at 298 K), we apply the linearized approach (e.g. Seinfeld and Pandis 2006), to predict the saturation ratio as a function of the mole solute on a insoluble core:

$$\ln\left(\frac{P_w(D_p)}{P^\circ}\right) = \frac{A}{D_p} - \frac{B}{(D_p^3 - d_u^3)}$$

In this equation A represents the Kelvin effect while B contains the solute effect:

$$A = \frac{4M_w\sigma_w}{RT\rho_w} \cong \frac{0.66}{T} \qquad (in\ \mu m)$$

$$B = \frac{6n_s M_w}{\pi \rho_w} \cong \frac{3.44 \times 10^{13} v m_s}{M_s} \qquad (in \ \mu m^3)$$

Herein $P_w(D_p)$ is the water vapor pressure over the droplet of diameter $D_p$; $P°$ is the water vapor pressure over a flat surface at the same $T$; $D_p$ is droplet diameter; $d_u$ is the diameter of insoluble particle fraction, in our case the $CaCO_3$ core; $M_w$ is water molecular weight; $\sigma_w$ is the air-water surface tension; $\rho_w$ is the water density; $T$ is in K; $R$ is the universal gas constant; $n_s$ is solute moles; $v$ is the dissociation degree, i.e. the number of ions resulting from the dissociation of one solute molecule; $m_s$ is solute mass per particle; $M_s$ is solute molecular weight.

As the solubility of $CaCO_3$ in water is very low, activation of $CaCO_3$ and other mineral dust components is often predicted by a water adsorption approach, wherein the solute term B in the Köhler equation above is replaced by a water adsorption term. The equations show application of the Frenkel Halsey Hill adsorption isoterme (FHH, Sorjamaa and Laaksonen, 2007, Kumar et al. 2009):

$$B = -A_{FHH} \cdot \theta^{-B_{FHH}}$$

Therein we approach the water coverage $\theta$ by (Sorjamaa and Laaksonen, 2007):

$$\theta = \frac{D_p - d_u}{2 \cdot 2.75 \cdot 10^{-4}}$$

Therein $A_{FHH}$ is a measure for the intermolecular interaction of water with the substrate surface whereas $B_{FHH}$ characterizes the intermolecular interaction within higher adsorbed water layers. Kumar et al. (2009) gave parameter for $CaCO_3$ ($A_{FHH}=0.25$ and $B_{FHH}=1.19$, FHH approach) and we will use this later to yield $SS_{crit}$ for bare $CaCO_3$ particles.

We applied Koehler theory to predict the CCN activation for the malonic acid coatings on the $CaCO_3$ particles at T = 293 K. We neglected the little dissolvable $CaCO_3$ and assumed that only malonic acid molecules contributed to the dynamic growth. The insoluble $CaCO_3$ core was set to $d_u$ = 0.1019 $\mu m$ as measured. We further assume that $CaCO_3$ doesn't react with malonic acid and that the surface

tension of the solution at activation is that of pure water. The dissociation constants $pkA_1$ and $pkA_2$ of malonic acid are 2.8 and 5.7 in water, respectively. So MA will partly dissociate in water and in the limit of infinite dilute solutions the dissociation degree $v$ (van' Hoff factor) will be three for full dissociation. The amount of malonic acid coating $m_s$ was taken from the AMS data in Table 1 (revised) in our manuscript and the mole MA solute was calculated applying the dissociation degree of $v = 3$ and the molecular mass of malonic acid of $M_s = 104.1$ g mol$^{-1}$. The resulting Koehler curves, equilibrium supersaturation (SS) over the solution droplet as a function of the wet diameter $D_p$, are shown in Figure S1. In Figure S1, the maximum of each SS curve is the critical supersaturation (theory $SS_{crit}$). In Table S1 and Figure 2S we compare the $SS_{crit}$ predicted by the Köhler approach (red) with the observed SScrit (black). Koehler theory overpredicts $SS_{crit}$ for thin coatings, but with increasing coating Koehler theory approaches the observed $SS_{crit}$ and the $SS_{crit}$ for a particle with 121 nm diameter composed of pure malonic acid is the limiting case (red filled circle).

In Figure S2 are also shown the $SS_{crit} = 1.49$ for bare $CaCO_3$ particles with $d_{dry} = 101.9$ nm, calculated from the observed $\kappa$'s in our study, and the range of $\kappa$'s given in the literature for wet generated particles (Tang et al. 2016).

From the Koehler results we derived the predicted water content of the particles at point of activation and we calculated molality and mass fraction of the solute in the solution at the point of activation. The molality at minimum and maximum malonic acid load of $3.3 \cdot 10^{-12}$ ug/particle and $610 \cdot 10^{-12}$ ug/particle were 0.006 mol kg$^{-1}$ and 0.0015 mol kg$^{-1}$, respectively. We used these values in the AIOMFAC model (Zuend etal. 2011) to calculate the deviation from ideality for the solution at point of activation for a flat solution. The both solutions are highly non-ideal with respect to the MA ($a_x = 0.4$), wherein MA was treated as solute with reference state infinite dilution (mole fraction $x_{solute} \rightarrow 0$). However this did not affect much the activity coefficient of water, which is essentially 1, water treated as solvent with reference state pure liquid (mole fraction $x_{water} = 1$). Moreover, in this concentration range, the surface tension of aqueous malonic acid solutions is about 70 dyn/cm, thus the nearly same as for water (Table S2). One should expect that Koehler theory would predict SS quite well under such conditions.

One could imagine that a little $CaCO_3$ will dissolve in water and in the presence of malonic acid, $CO_2$ should be driven out, and a solution of CaMalonate could be formed. CaMalonate is still slightly soluble in water (0.365 g/100 g water, at 293 K, Linke and Seidell, 1958), $v$ for CaMalonate is 2. Solubility of CaMalonate converts into about 0.02 mol/kg water, would thus be in a range where the malonate would be fully dissolve. However, because of $v = 2$ for CaMalonate while $v = 3$ for the pure malonic acid, formation of CaMalonate will decrease the solute effect.

To bring Köhler theory in agreement with the observation for the thinnest coating, *more* solute entities would be required. Thus, disagreement cannot be caused by an ill determined van't Hoff factor as we used already maximum $v = 3$ and reducing $v$ will increase the deviation. (Note, that recent observations point to the importance of the surface effect by organic surface films over the solute effect for water soluble inorganics in presence of organics, including malonic acid (Ruehl et al., 2016). A lower surface tension will bring Koehler prediction and observation punctually in better agreement and still allow for smaller van't Hoff factors. As an example, at point of activation a surface tension of 55% of $\sigma_w$ and a van't Hoff factor of one will bring Koehler theory and observation in agreement for the thinnest coating. However, a surface tension 55% of $\sigma_w$ will cause disagreement for the thickest coating, because the solute term gains in importance. Probably, the findings for the mixed solutions of malonic acid and water soluble ammonium sulfate are not directly transferable to our system with insoluble inorganic core, where we expect dilute aqueous solutions of 0.006 mol/kg of malonic acid at the activation point. At such concentrations malonic acid does not reduce $\sigma_w$, moreover in the study of Ruehl et al. (2016) malonic acid was one of the more Koehler kappa behaving organics.)

If we turn back to Koehler theory with focus on the solute term, about 4-5 times the measured MA would be needed for the thinnest MA coating to bring prediction and observation in agreement. It cannot be due to a simple calibration error of 4-5, as that would apply to all coating amounts and would lead to mismatches at the thicker coatings. Moreover, a postulated missing mass of $14\text{-}17.5 \cdot 10^{-12}$ ug/particle will be detectable by AMS.

CaCO$_3$ aerosol in this study was generated by spraying saturated Ca(HCO$_3$)$_2$ solutions. The solubility of Ca(HCO$_3$)$_2$ in water is with 16.6 g/100g water sufficient that about 1% Ca(HCO$_3$)$_2$ would provide the missing amount of solute. However, we dry and temper the aerosol at 300°C and Ca(HCO$_3$)$_2$ in dry state is thermodynamically instable (Zhao et al. 2010). As described in the manuscript, our wet generated CaCO$_3$ particles are more hygroscopic as Calcite and dry generated CaCO$_3$ particles. Larger $\kappa$ for wet generated compared to dry generated CaCO$_3$ particles is a common phenomenon (e.g. Tang et al. 2016) and possibly related to the formation of Ca(OH)(HCO$_3$) structures at the surface (see below).

In Figure S2 we show the prediction of SS$_{crit}$ 1.52% for activating CaCO$_3$ by the Kelvin/FHH theory with the CaCO$_3$ parameters taken from Kumar et al. (2009). SS$_{crit}$ for our bare CaCO$_3$ particles is 1.49% and the lower SS$_{crit}$ should be due to a more adsorptive surface, e.g. the presence of Ca(OH)(HCO$_3$) structures. According to classical Koehler theory the equivalent of $8.5 \cdot 10^{-20}$ moles of dissolvable entities would be needed to explain a drop from 1.52% to 1.3% supersaturation, which is only ¼ of the moles MA in the thinnest MA coating. Therefore, whatever makes our CaCO$_3$ particles wettable is not sufficient - in terms of Koehler theory - to explain the low SScrit of 0.9 % at the thinnest MA coating.

We estimate monolayer coverage by MA at 2-3% volume fraction (vf), this would be achieved between the third and fourth MA mass load of $13 \cdot 10^{-12}$ - $38 \cdot 10^{-12}$ ug per particle, observed vf 1.5-4.1%. Thus a sub-monolayer coating of $3.3 \cdot 10^{-12}$ ug MA per particle caused a drop of SS$_{crit}$ from 1.49 to 0.9 and increased kappa from 0.0028 to 0.012. Therefore we conclude that CaCO$_3$/MA coatings show a *non-Koehler* behavior at thin coatings and *approach* Koehler behavior with increasing MA load. This means there must be specific interactions between MA and the CaCO$_3$ surface which eases water adsorption and CCN activation.

Ca(OH)(HCO$_3$) structure exists commonly on the surface of CaCO$_3$ whenever the CaCO$_3$ surface has been exposed to gaseous water or liquid water. Ca(OH)(HCO$_3$) exist even at high vacuum condition and act as hydrophilic sites on the surface of CaCO$_3$ making the CaCO$_3$ surface more hydrophilic (Stipp, 1999; Stipp and Hochella, 1991; Neagle and Rochester, 1990). CaCO$_3$ aerosol in our study was generated by

spraying a $Ca(HCO_3)_2$ solution, so $Ca(OH)(HCO_3)$ structures likely exist on the particle surface. When $CaCO_3$ particles are coated by malonic acid (or oleic acid) the hydrophilic sides can serve as polar surface active sites for accommodation of the acids. In case of MA there is a second carboxylic group which still could support the adsorption of water films.

If we think in terms of Kelvin/FHH theory to explain the observed low $SS_{crit}$ for thin MA coatings, these coatings must have a net stronger interaction with water (higher $A_{FHH}$) and/or stronger interaction between the adsorbed water layers (lower $B_{FHH}$) than bare $CaCO_3$. If coatings become thicker the Koehler solute effect starts increasingly to contribute and eventually controls the CCN activation. But our data are not sufficient to determine $A_{FHH}$ and $B_{FHH}$. (The only system in the literature which comes close - in a far sense - is CaOxalate Monohydrate, with $A_{FHH} = 0.57$ of and $B_{FHH} = 0.88$ (Kumar et al., 2009). Plugin in these parameters will lead to $SS_{crit} = 0.53\%$ commensurable with our observed value of 0.56% for $13 \cdot 10^{-12}$ ug MA coating which represents an organic^vf of 1.6%, thus is close to monolayer.)

Oleic acid (OA) is not soluble in water. Pure oleic acid particles up to 333 nm do not activate at 0.87% SS. This data sets the upper limit of $\kappa$ for pure oleic acid to < 0.0005. With thin coatings of OA, CCN activity is smaller than bare $CaCO_3$, however with thick coatings it becomes larger. This is not a classical Koehler behavior. Therefore adsorption e.g. described by Kelvin/FHH theory is a better approach to illustrate what could happen on coating with oleic acid.

Note that we are neither able to perform theoretical calculations nor is our data sufficient to determine e.g. FHH parameters. Both are clearly beyond this experimental work.

We again refer to $Ca(OH)(HCO_3)$ structures at the surface which offer polar surface sites to bind the hydrophilic ends (the carboxyl groups) of the oleic acid molecules. Different to malonic acid, only hydrophobic ends of oleic acid molecules (the hydrocarbon chains) are then exposed on the particle surface hence increase the hydrophobicity of the particle surface. Such a formation of a hydrophobic layer

should be occurring until all polar sites are occupied or a monolayer coverage - maybe in form of a self-assembled layer - is reached. This can hinder the uptake of water, and in terms of Kelvin/FHH theory will lower $A_{FHH}$ and/or likely increase $B_{FHH}$. The formation of a monolayer of OA on black carbon particles with the polar groups pointing outwards was postulated by Dalirian et al. (2017), which lead to increased activation of the black carbon particles. Thus, they observed a similar effect of layer formation, but with switched polarity.

When $CaCO_3$ particles are coated with more oleic acid (more than one monolayer) the first layer of oleic acid still combines with $CaCO_3$ surface. We suppose that a portion of the carboxyl groups of oleic acid molecules, which are not in the first layer, will be exposed on the particle surface in analogy to the formation of lipid bilayers, e.g. cells. The particle surface then becomes more hydrophilic and in terms of Kelvin/FHH theory the interaction of water with the OA surface becomes stronger, $A_{FHH}$ increases and likely also the interaction between the higher water layers ($B_{FHH}$ decreases). From this point of view water adsorption should become similar to thin malonic acid layers. In addition, when droplets form, oleic acid will transfer to the surface of the droplets and lower the surface tension of the solution (the surface tension of oleic acid is 0.033 J m$^{-2}$, which is much lower than that of pure water of 0.072 J m$^{-2}$). Thus, the activation of OA coated particles is probably a complex interaction between formation of specific hydrophobic layers and more hydrophilic multilayers, surface tension and for the largest coatings amount size effects. As shown in Figure S2, SScrit for OA is lower than for thin malonic acid coatings, probably because of the surface tension effect, but higher than for thick MA coatings, because of the missing solute effect.

We will the discussion above implement in parts in the manuscript. This passages are highlighted in yellow in the revised manuscript.

*C2. Focusing on the coating device temperature in the results section is a bit confusing and I am not convinced about the reproducibility of the experiment, given*

*that the experimental cases are identified mainly by the temperature of the coating device. Focusing more on presenting the mass of organics coated on the particles would improve the paper significantly. Does the coating thickness depend on the original particle size? Can this give more information towards understanding the observations?*

**Response:**

First of all we would like to clarify that we did not meant that coating thickness depends only on temperature. Our intension was to use the coating temperature to classify our experiments, because the organic mass itself has some uncertainty, related to limitations of the measurements method. It means that we are convinced that for a given particle size (distribution) and fixed experimental conditions (flows, residence time etc.) will indeed lead to reproducible amounts of coatings for a given temperature.

In Table 1 (revised) in our manuscript we showed the organic mass per particle data. We will focus on the organic mass and organic volume fraction in the revision of our manuscript. Due to the Kelvin Effect, the amount of organic acid coating depends somewhat on the particle size of the core. In our study, we selected $CaCO_3$ particles at a diameter of 101.8nm and detailed discussion focus on the impact of coating thickness on the CCN activity of 101.8nm $CaCO_3$ particles. However, as shown in Figures   polydisperse particles with a distribution of particle size and coatings bhave similar.

*C3. In the motivation of the chosen substances: is oleic acid really a good choice for a surface-active compound, considering its low solubility?*

**Response:**

Yes, we think that the two acids are good choices as the water solubility of the two organic acids is complementary; it is high for malonic acid while it is very low for oleic acid, and we speculated in our planning that oleic acid would form kind of self-assembled layers. Surface active compounds do not need to be water soluble. Moreover, oleic acid is observed in the atmosphere.

*C4. On line 179, the residence time in the flow tube is given. However in a laminar flow, the residence time varies with the position, i.e. whether the air passes at the center or near the walls. Is this effect considered and may it influence the results?*

**Response:**

The flow tube was supposed to act as delay tube. The residence time of 23.7 s for the particles in the flow tube is the average residence time. This compares to the average residence time of only about 5.8 s for the particles in the coating device (including the water cooler). Thus average residence time in the flow tube is about 4 times of that in the coating device. 1.) In the way we sampled aerosol from the flow tube, the aerosol contains flow both from the center and near the walls. 2.) increasing of residence time (by 23.7 s) had no significant impact on CCN activity at all conditions, thus fact that particles travelling in center and particles travelling near the wall was not important.

*C5. Lines 180 to 185: At first reading it sounds as if the relative humidities given are at 80 ºC. However, in the case of 47% RH, the air would then be strongly supersaturated at 25 ºC. Please clarify the temperature at which the RH is given. Also, if 47% RH refers to 25 ºC, the RH at 80 ºC will still be very low, just a few percent. How does this influence the experiments?*

**Response:**

We thank the referee for the comment as our description maybe indeed misleading. RH experiments were performed at 60°C(MA) and 80°C(OA) coating temperature. The $N_2$ stream was passed through a bubbling device filled with Milli-Q water to saturate the $N_2$ stream and RH was measured to be >90% at T =25 ºC (room temperature). The RH of the aerosol flow before entering the coating device is measured to be <10% at T =25 ºC. The RH of the mixed stream should be about 50% at 25 ºC, since the flow rate of the $N_2$ stream and the aerosol flow is the same. This corresponds to water vapor pressure of 1500 Pa. So, within the coating device at 60°C the RH will be >7% (MA experiment) and at 80°C >3% (OA experiment), still about an order of magnitude higher than the RH, when the stream is not humidified. In fact it will be a somewhat higher as the gas-phase may not reach the bath temperature which primarily served to warm up the coating agent and control its vapor pressure.

During passing through the cooler, RH increases to 47% at 25°C at the exit of the cooler.

We clarified this in the manuscript.

*C6. In lines 228 to 231 proportionalities between AMS signals and organic particle mass and volume are described. Are they observations or assumptions? Also, determination of the amount of organic coating: If I read the table 1 correctly, and make some calculation, the change in diameter and the change in organic mass, do not agree. Is this correct? Why is that?*

**Response:**

Yes, AMS determined mass and observed volume increase should be linearly related. This is the working hypothesis. There are apparent deviations from strict linearity, mainly because of limitations in determining the volume increase through coating, as the formed layers are thin, and growth is smaller than the bin width of the SMPS. The size resolution of the SMPS was set to maximum of 64 channels per decade.

AMS is obviously able to detect the smallest coating amounts, as described in the manuscript, but standard relative ionization efficiency (RIE = 1.4) for organics, which can be well applied to atmospheric systems, fails in our case as it underestimates the volume observed by SMPS for the thickest coating by more than a factor of two (applying the macroscopic densities). It is well known that application of average RIE for organics often fails for specific single compounds.

Therefore it seemed to us, a reasonable way out could be the procedure which we applied and described in the experimental section. We assume the particles with the thickest coating are spherical, coatings are bulk like, and can be measured by SMPS with sufficient accuracy. Then we calibrated the largest *m/z* observed by AMS as marker for the thickest coating, and apply the marker calibration to the thinner coatings. As can be seen from the STDEV the reproducibility is quite good but there are systematic errors. The bin width "uncertainty" in the range of diameters from 101.8 to 120.9 nm is 3.9 and 4.3 nm, respectively. At coating thicknesses of about 5 - 20 nm this introduces uncertainties of 32 to 12% already. If one recalculates the

expected diameter for the coated particle from core + coating volume

$$D_{coat} = \sqrt[3]{D_{core} \cdot \frac{6}{\pi} \cdot \frac{m_{org}}{\rho_{org}}}$$

the derived diameters $D_{coat}$ fall in the bin-width of the channel noted in Table 1 (old) as mode diameter. As a consequence within these uncertainties the relation between SMPS volume and AMS mass agree. (These uncertainties of the coating mass was the reason why we use the operational coating temperature to classify our experiments).

We will show the relation between AMS data and SMPS data in Figure S1 in the supplement. See response to next comment.

*C7. Line 234 to 235. Have you tried to account for mode shift within the bin, by fitting a function to the distribution?*

**Response:**

Thanks for the suggestion. We now fitted the maxima of the SMPS number distribution by a lognormal function (considering 5 to 9 neighbor bins centered around the mode), and took the maximum of the fits as mode diameter. This led indeed to changes. The most significant changes were that for the thickest coating the fitted maxima shifted from 121.9 to 121 nm and from 121.9 to 124.3 nm for MA and OA respectively. This led to different calibration factors and thus to different coating amounts for the thinner coatings. The effects were -5% and +11% for OA and MA, respectively. By interpolating the mode position the linearity between SMPS volume and AMS also improved as indicated in Figure S3. In the Figure S3 also the mass range over the bin width is given as x error bars. (The error bars are different for MA and OA as the density $\rho_{org}$ entered the calculation.) One can see that mass calculated with nominal mode position (old Table 1) is centered over the bin width bars. One can also recognize how much the mass calculated from the interpolated mode diameters

(Table 1(revised)) were shifted inside the bin width bar.

We will use the revised data in the revised manuscript. See previous comment.

*C8. Line 293: You say "and our kappa value is somewhat higher but still in this range". Isn't it just in the range?*

**Response:**

Yes, it is just in the range. We will correct in the manuscript and will mention that larger kappas for wet generated $CaCO_3$ has been found before.

*C9. Line 298: You say "The increased kappa value of 0.0008". Shouldn't it be "The increase in kappa value of 0.0008"?*

**Response:**

Yes, it should be "The increase in kappa value of 0.0008". We will correct in the manuscript.

*C10. Line 323: What do you mean by a "significant amount" of coating? Is a coating of less than 2 nm insignificant? In all aspects?*

**Response:**

We will reformulate that section.

*C11. Line 323-325: Unclear expression. Please specify that the sizing according to mobility diameter is referred to (if that is the case) and specify the temperature limit, instead of saying "a certain value".*

**Response:**

We will reformulate that section.

*C12. Line 371: Why does the activated fraction level off at 83%? Did you make an intercalibration between numbers in the CCNC and CPC for other compounds?*

**Response:**

The activation efficiency (i.e., the activated fraction when aerosol particles are completely activated) was 83% for our CCN instrument, determined using 150 nm $(NH_4)_2SO_4$ particles at SS=0.85%. In Figure 4 and Figure 6 in our manuscript, reaching the plateau of about 83% means that the aerosol particles were completely activated. Activation efficiency around 80% are typical compare the study of Abbatt et al. (2005). We prefer not to correct for better looks.

*C13. In section 3.2 there is a lot of repetition, I think. Please see if the text can be made more efficient.*

**Response:**

We will try to make the text more efficient and avoid repetition.

*C14. Line 413, the sentence starting with "This suggestion is supported by very low CCN activity of pure oleic acid...." Is this based on some calculations? Please describe. Line 421 to 423: But the low surface tension does not seem to help the pure oleic acid particles to activate, according to your measurements as well as Kumar et al. (2003) and Broekhuizen et al. (2004). Also, the first "of" on line 422 should be removed.*

**Response:**

Our CCN activity measurements show that oleic acid particles up to 333 nm do not activate at 0.87% SS, thus pure oleic acid particles have very low CCN activity ($\kappa$ < 0.0005). The research of Kumar et al. (2003) and Broekhuizen et al. (2004) also showed pure oleic acid particles have very low CCN activity. In pure oleic acid particles micelle like structures are formed, with the hydrophilic ends (the carboxyl groups) of the oleic acid molecules tend to combine together by hydrogen bonds and the hydrophobic tail (the hydrocarbon chains) are exposed at the outside (Iwahashi et al., 2000; Garland et al., 2008). The arrangement of oleic acid molecules in pure oleic

acid particles should be similar to that in pure oleic acid liquid. This can increase the hydrophobicity of the particle surface and hinder the uptake of water. Even low surface tension cannot help the pure oleic acid particles to activate. And oleic acid is highly insoluble in water.

We will remove the first "of" on line 422.

**C15.** *Figure 6: The fitted curves for 30-60 ºC do not seem to follow the data very well. Why? Have you accounted for doubly charged particles? In my opinion there is a tendency to a two step-function, the first step reaching to CCN/CN of 0.15-0.2.*

**Response:**

We can deduct the presence of doubly charged particles in the CN measurement, but we cannot deduct the activated doubly charged particles from CCN. One method for solving this problem is provided by Sullivan et al. (2009). In their paper they fitted the data from the "multiply-charged plateau" to the "completely-activated plateau" when the data had a clear "multiply-charged plateau". But in our data, we don't have a clear "multiply-charged plateau", so we fitted from the beginning to the "completely-activated plateau". This will lead to a systematic underestimate of the $SS_{crit}$. The underestimate is the largest when the malonic acid mass is the smallest (30 ºC coating temperature), and we underestimate $SS_{crit}$ by only about 0.02% (difference between fitting from the first point and fitting from the third point (multiply-charged plateau). When the malonic acid mass is larger, the underestimate will be smaller. At the largest two malonic acid mass, the underestimate can be neglected.

We will also make that clearer in the revision in our manuscript.

[revised manuscript text omitted]

**Figure S2:** Comparison of SS$_{crit}$ predicted by Koehler and Koehler/FHH theory with observations. The red circles are predictions by the Köhler theory for aqueous MA solutions assuming full dissociation, the black points present the observation. The red filled circle represents the Koehler prediction of SScrit for 121.0 nm particles made of pure malonic acid. The horizontal lines give reference values for the bare CaCO$_3$ particles as calculated from our observed $\kappa$ (black) and predicted by Koehler/FFH theory (blue). Light grey area between the thin dashed black lines indicates the range of SS$_{crit}$ for 101.9 nm particles calculated from the range of $\kappa$ in literature for wet generated CaCO$_3$ particles. Green diamonds show observed SS$_{crit}$ for the two thickest OA coatings.

[Figure]

**Figure S3:** Linear relation between organic mass of the coating (AMS) and the organic coating mass calculated from the SMPS measurements for oleic acid coatings (green) and maleic acid coatings (black). For full circles the organic mass was calculated form the interpolated mode diameters (new). For interpolation we fitted a lognormal function in the range of the nominal mode diameter (considering 5-7 bins). The open circles are the data from the manuscript, where the nominal mode diameters were used to calculate the organic mass expected from SMPS measurements. Error bars in y give the reproducibility of the AMS measurement, error bars in x indicate the mass range (volume range) spanned by the SMPS bin width for the respective nominal mode diameter that entered the mass calculation. Note that the highest values must be the same as they were used to calibrate AMS organic mass vs SMPS volume.

**Table S1.** Theory and experimental $SS_{crit}$ and κ values of $CaCO_3$ particles with different mass of MA coating.

| Dry diam. (nm) | MA per particle ($\times 10^{-12}$ μg) | Theory SScrit (%) | Experimental SScrit (%) | Theory κ | Experimental κ |
|---|---|---|---|---|---|
| 102.0 | 3.3±0.3 | 1.44 | 0.90 | 0.0024 | 0.0123±0.0005 |
| 102.1 | 6.8±1.2 | 1.20 | 0.70 | 0.0053 | 0.0231±0.0008 |
| 102.2 | 13±1.8 | 0.97 | 0.56 | 0.0107 | 0.0380±0.0012 |
| 102.7 | 38±1.6 | 0.63 | 0.35 | 0.0315 | 0.1056±0.0023 |
| 107.8 | 160±8.1 | 0.31 | 0.24 | 0.1110 | 0.1813±0.0031 |
| 121.0 | 610±24 | 0.16 | 0.16 | 0.3001 | 0.3001±0.0062 |

**Table 2.** Properties of investigated compounds.

| Name | Formula | Molecular weight (g mol$^{-1}$) | Density (g cm$^{-3}$) | Solubility (g/100 g water, at 298K) | Surface tension (dyn/cm) |
|---|---|---|---|---|---|
| Calcite | $CaCO_3$ | 100.1 | 2.71 | 0.00058 | -[a] |
| Malonic acid | $C_3H_4O_4$ | 104.1 | 1.62 | 62 | 69[b] |
| Oleic acid | $C_{18}H_{34}O_2$ | 282.5 | 0.89 | Very low | 33[c] |

Values are taken from: Handbook of Chemistry and Physics, the 82nd Edition; Handbook of aqueous solubility data, Samuel H. Yalkowsky and Yan He, 2003; Chumpitaz et al., 1999; Hyvarinen et al., 2006.

a: no data

b: Surface tension of 0.01 mole fraction malonic acid aqueous solution at 298K

c: Surface tension of pure oleic acid at 293K

Chumpitaz, L. D. A., Coutinho, L. F., and Meirelles, A. J. A.: Surface tension of fatty acids and triglycerides, J. Am. Oil Chem. Soc., 76, 379-382, 10.1007/s11746-999-0245-6, 1999.

Hyvarinen, A. R.; Lihavainen, H.; Gaman, A.; Vairila, L.; Ojala, H.; Kulmala, M.; Viisanen, Y., Surface tensions and densities of oxalic, malonic, succinic, maleic, malic, and cis-pinonic acids. Journal of Chemical and Engineering Data 2006, 51 (1), 255-260.

---

## Author Comment (AC2) · 27 Mar 2018

We thank the referee for the constructive comments. The responses are attched as pdf-file, with an extra figure.

Please also note the supplement to this comment:
https://www.atmos-chem-phys-discuss.net/acp-2017-897/acp-2017-897-AC2-supplement.pdf

[Figure]

**Figure S1:** Comparison of $SS_{crit}$ predicted by Koehler and Koehler/FHH theory with observations. The red circles are predictions by the Köhler theory for aqueous MA solutions assuming full dissociation, the black points present the observation. The red filled circle represents the Koehler prediction of SScrit for 121.0 nm particles made of pure malonic acid. The horizontal lines give reference values for the bare $CaCO_3$ particles as calculated from our observed $\kappa$ (black) and predicted by Koehler/FFH theory (blue). Light grey area between the thin dashed black lines indicates the range of $SS_{crit}$ for 101.9 nm particles calculated from the range of $\kappa$ in literature for wet generated $CaCO_3$ particles. Green diamonds show observed $SS_{crit}$ for the two thickest OA coatings.

**Fig. 1.**

**Supplement:**

**Cloud Condensation Nuclei Activity of CaCO₃ Particles with Oleic Acid and Malonic Acid Coatings**

**Mingjin Wang[1,2], Tong Zhu[1*], Defeng Zhao[2], Florian Rubach[2,3], Andreas Wahner[2], Astrid Kiendler-Scharr[2], and Thomas F. Mentel[2*]**

**A figure for illustration of the responses is given in an extra file**

*C1: Overall the paper is nicely illustrated but it contains some repetitive text blocs, especially in the result section, that could be avoided in order to ease the reading of this manuscript (and reinforcing also its content).*

**Response:**

We thank the referee for the positive comment. We will optimize the text of our manuscript to avoid repetition and to make the text more efficient and easier to read.

*C2: While the argument of a bilayer of oleic acid does make sense, I'm still puzzled by the fact that the CCN properties are apparently higher for a coated particle compared to a pure oleic acid particle. I would have simply assumed that once thick enough, the water probing the surface does not see the core CaCO₃ particle (over the time scale of these experiments). In this situation, the pure oleic acid particle would exhibit a kind of upper limit for water adsorption and droplet activation. Maybe the authors could comment more on that, and maybe add the pure oleic acid data on their figures (this would ease the comparison with both systems).*

**Response:**

Our CCN activity measurement showed that pure oleic acid particles up to 333 nm did not activate at 0.87% SS; this sets an upper limit for CCN activity of oleic acid particles ($\kappa < 0.0005$), in agreement with Kumar et al. (2003) and Broekhuizen et al. (2004). In liquid state oleic acid (OA) forms micelle like structures, the hydrophilic

ends (the carboxyl groups) of oleic acid molecules tend to combine together by hydrogen bonds and the hydrophobic tails (the hydrocarbon chains) are exposed at the outside (Iwahashi et al., 2000; Garland et al., 2008). The arrangement of oleic acid molecules in pure oleic acid particles should be similar. Hydrophobic tails facing outwards can explain the hydrophobicity of the particle surface and the hindrance of the uptake of water, making the CCN activity of pure oleic acid particles very low. Dalirian et al. (2017) studied OA coatings on black carbon (BC). They also found *no* activation of pure OA, but droplet activation for OA coated BC particles at $SS_{crit}$ lower than that of BC and pure OA. Their particles were larger than ours, though, but also more heavily coated, i.e. more OA like. So, better droplet activation than for pure OA seems to be possible in coated systems.

We assume that $Ca(OH)(HCO_3)$ structures act as hydrophilic sites on the surface of $CaCO_3$ (Kuriyavar et al., 2000; Stipp, 1999; Stipp and Hochella, 1991; Neagle and Rochester, 1990). Garland et al. (2008) suggested that OA at sub-monolayer coverage form self-associated islands rather than uniformly covering the surfaces, and OA molecules were oriented vertically on both hydrophobic and hydrophilic surfaces with the hydrocarbon chains of oleic acid molecules facing away from the surface. This is in support of our working hypothesis: the formation of a hydrophobic surface film.

When $CaCO_3$ particles are coated with only a small amount of oleic acid (less than one monolayer), the hydrophilic ends (the carboxyl groups) of oleic acid molecules will combine with the hydrophilic sites (-OH and - H $CO_3$ groups) on $CaCO_3$ surface by hydrogen bonds. We conclude that all hygroscopic sides on the $CaCO_3$ surface are covered somewhere between 50°C and 60°C coating temperature, i.e. between $14 \cdot 10^{-12}$ and $23 \cdot 10^{-12}$ ug OA mass per particle, as here the trend turns and droplet activation increase again. This would place the monolayer coverage above 3%, organic volume fraction. According to the measurements and calculations of the length of oleic acid molecule, the thickness of oleic acid sub-monolayer on solid surfaces, and the thickness of deuterated oleic acid monolayers at the air-water interface (Garland et al., 2008; King et at., 2009; Iwahashi et al., 2000), we determined 2.3 nm as the likely thickness of oleic acid monolayer on $CaCO_3$ particles, accordingly a monolayer would be achieved at about 12-13% organic volume fraction. As a consequence the formation of the "hydrophilic bilayer" starts at sub-monolayer

coverage in accordance with island formation observed by Garland et al. (2008).

For $CaCO_3$ particles coated with more OA (thicker than one monolayer, at 70 and 80 °C coating temperatures), OA in the first layer still combines with the $CaCO_3$ surface by hydrogen bonds. We suppose that a portion of the carboxyl groups of oleic acid molecules, which are not in the first layer, will be exposed to the particle surface, in analogy to the formation of lipid bilayers, e.g. in cells, though the structure of this part of oleic acid is not known. The particle surface then becomes more hydrophilic. Activation of wettable, insoluble material can be described by the Kelvin term and a water absorption term, e.g. Frenkel Halsey Hill isotherm (Sorjamaa and Laaksonen, 2007, Kumar et al. 2009).

When carboxylic groups of OA are exposed at the surface, the interaction of water with the OA layer becomes stronger. In terms of the Kelvin/FHH approach, the surface water interaction becomes stronger and $A_{FHH}$ increases and likely also the interaction between the higher water layers ($B_{FHH}$ decreases). From this point of view water adsorption by the "OA bilayer" should become similar to thin malonic acid layers. In addition, when droplets form, oleic acid will transfer to the surface of the droplets and lower the surface tension of the solution (the surface tension of oleic acid is $0.033$ J m$^{-2}$, which is much lower than that of pure water of $0.072$ J m$^{-2}$). Thus, the activation of OA coated particles is probably a complex interaction between formation of specific hydrophobic layers and more hydrophilic multilayers, surface tension effects and for the largest coating amounts, simple size effects. As shown in Figure S1, $SS_{crit}$ for OA is lower than for thin malonic acid coatings, probably because of the surface tension effect, but higher than for thick MA coatings, because of the missing solute effect.

We will implement parts of this discussion in the manuscript, highlighted in yellow.

*C3: If my reading is correct, the CaCO₃ particles with a thick coating do exhibit better CCN properties than the pure oleic acid particles (if my reading is incorrect, this would highlight that an in-depth editing would be beneficial for the reader). How can you explain such a fact? Is the CaCO₃ surface leading to some kind of ordering of*

*the adsorbed organic acid (which might not be observed in the pure homogeneous organic particle) leading effectively to the above mentioned bilayer structure?*

**Response:**

Your reading is correct. Note, the oleic acid coatings achieved at 70° or 80°C coating temperature are still thin. The largest coating thickness is about 10 nm at an organic volume fraction of 44%, which corresponds to about 4 monolayers of oleic acid (assuming the thickness of oleic acid monolayer on $CaCO_3$ particles is about 2.3 nm). To our knowledge, no previous study has investigated the structure of oleic acid coating on particle or solid surface thicker than one monolayer.

In our understanding the arrangement of oleic acid molecules in the thin coatings will be influenced by the $CaCO_3$ core (the polar, hydrophilic sides) and can thus be different from the arrangement of oleic acid molecules in pure oleic acid particles. Water can probably adsorb at the carboxylic groups facing outward ("bilayer" type structure) and diffuse through the thin oleic acid coatings. It may form an adsorbed water phase near the $CaCO_3$ surface. This could push the oleic acid out to act as surfactant which lowers the Kelvin term. Such a process should also happen in pure oleic acid particles. Because of the presence of $CaCO_3$ core the SS to achieve is lower than for pure OA.

We will explicitly refer to the lower $\kappa$ of pure oleic acid in relation to higher $\kappa$ observed for the oleic acid coated $CaCO_3$.

*C4: Is the temperature in the coating device leading to some kind of ordering (for instance, by increase surface mobility before an ordering when cooling down)? Also, oleic acid has only a moderate thermal stability as it decomposes at higher temperature (typically at temperature at a factor 2 higher than those used here), potentially via an epoxide pathway in presence of air (and enhanced by light or metallic traces). Could traces of oxidized products nevertheless affect the composition of the coating at 80 ℃ already? If so, then this effect should increase with time. Did the authors observed any variation with time of kappa at 80 ℃? Did the authors tried*

*to have thicker coatings at lower temperature by changing the gas flow conditions in their coating device?*

**Response:**

We agree with the referee that the temperature in the coating device can increase the surface mobility of OA molecules, which would help to optimize the layering arrangement (ordering) of adsorbed oleic acid molecules on $CaCO_3$ surface.

We can exclude oxidation of OA since we used high-purity $N_2$ (Linde LiPur 6.0, purity 99.9999%, Linde AG, Munich, Germany) and we did not have much light in our system because both the coating device and the flow tube were covered with light tight materials (black polyurethane foam). Metallic traces might exist in our system, though, as we used some stainless steel tubes in our system.

The only way in which we modified the time between coating process and particle detection was provided by a flow tube to increase the average residence time from 5.8 to 23.7 s. Longer residence time (by 23.7 s) had no significant impact on CCN activity for both oleic acid coated particles and malonic acid coated particles at both dry and 47% RH conditions, probably because the coating process was already completed in the coating device and no further reactions occurred in the flow tube.

At the beginning of our experimental study, we modified the gas flow condition in the coating device in order to choose an appropriate gas flow condition. But we only measured the particle size when we changed the gas flow condition. After finding the appropriate gas flow conditions, we fixed the gas flow condition and did not change it anymore.

*C5: Line 238. This sentence is unclear.*

**Response:**

This sentence should be: In the discussion we also use the coating mass per particle to give a rationale, which is easier to imagine, to the finding and classification.

*C6: Line 249: where a polydisperse*

**Response:**

This sentence should be: In cases where a polydisperse aerosol was coated …

*C7: Line 296: remained at*

**Response:**

This sentence should be: The κ value remained at 0.0028 ± 0.0001 at 60 ℃ and increased to 0.0036 ± 0.0001 at 80 ℃.

And we also made the revision in our manuscript according to your comments.

**References**

Broekhuizen, K. E., Thornberry, T., Kumar, P. P., and Abbatt, J. P. D., Formation of cloud condensation nuclei by oxidative processing: Unsaturated fatty acids, J. Geophys. Res.-Atmos., 109, D24206, 10.1029/2004jd005298, 2004.

Dalirian, M., Ylisirniö, A., Buchholz, A., Schlesinger, D., Ström, J., Virtanen, A., and Riipinen, I.: Cloud droplet activation of black carbon particles coated with organic compounds of varying solubility, Atmos. Chem. Phys. Discuss., 2017, 1-25, 10.5194/acp-2017-1084, 2017.

Garland, E. R.; Rosen, E. P.; Clarke, L. I.; Baer, T., Structure of submonolayer oleic acid coverages on inorganic aerosol particles: evidence of island formation. Phys. Chem. Chem. Phys. 2008, 10 (21), 3156-3161.

Iwahashi, M.; Kasahara, Y.; Matsuzawa, H.; Yagi, K.; Nomura, K.; Terauchi, H.; Ozaki, Y.; Suzuki, M., Self-diffusion, dynamical molecular conformation, and liquid structures of n-saturated and unsaturated fatty acids. J. Phys. Chem. B 2000, 104 (26), 6186-6194.

King, M. D.; Rennie, A. R.; Thompson, K. C.; Fisher, F. N.; Dong, C. C.; Thomas, R. K.; Pfrang, C.; Hughes, A. V., Oxidation of oleic acid at the air-water interface and its

potential effects on cloud critical supersaturations. Phys. Chem. Chem. Phys. 2009, 11 (35), 7699-7707.

Koehler, H., The nucleus in and the growth of hygroscopic droplets. Trans. Faraday Soc. 1936, 32 (2), 1152-1161.

Kumar, P. P., Broekhuizen, K., and Abbatt, J. P. D., Organic acids as cloud condensation nuclei: Laboratory studies of highly soluble and insoluble species, Atmos. Chem. Phys., 3, 509-520, 2003.

Kumar, P., Nenes, A., and Sokolik, I. N.: Importance of adsorption for CCN activity and hygroscopic properties of mineral dust aerosol, Geophys. Res. Lett., 36, 10.1029/2009gl040827, 2009.

Sorjamaa, R., and Laaksonen, A.: The effect of $H_2O$ adsorption on cloud drop activation of insoluble particles: a theoretical framework, Atmos. Chem. Phys., 7, 6175-6180, 10.5194/acp-7-6175-2007, 2007.

Kuriyavar, S. I., Vetrivel, R., Hegde, S. G., Ramaswamy, A. V., Chakrabarty, D., Mahapatra, S. Insights into the formation of hydroxyl ions in calcium carbonate: temperature dependent FTIR and molecular modelling studies. Journal of Materials Chemistry, 2000, 10 (8): 1835-1840.

Neagle, W., Rochester, C. H., Infrared study of the adsorption of water and ammonia on calcium carbonate. Journal of the Chemical Society, Faraday Transactions 1990, 86 (1), 181-183.

Stipp, S. L. S., Toward a conceptual model of the calcite surface: hydration, hydrolysis, and surface potential. Geochimica et Cosmochimica Acta 1999, 63 (19-20), 3121.

Stipp, S. L., Hochella Jr, M. F., Structure and bonding environments at the calcite surface as observed with X-ray photoelectron spectroscopy (XPS) and low energy electron diffraction (LEED). Geochimica et Cosmochimica Acta 1991, 55 (6), 1723-1736.

---

## Author Response (AR2)

Dear editor,

we thank again the reviewers.

We addressed the technical comments and also removed some typos in the manuscript and in the supplement.

With best regards!

Thomas Mentel